# Mitochondrial Complex I and ROS control neuromuscular function through opposing pre- and postsynaptic mechanisms

Bhagaban Mallik[1], Sajad A. Bhat[2], Xinnan Wang[2], C. Andrew Frank[1]*

1 Department of Anatomy and Cell Biology, University of Iowa Carver College of Medicine, Iowa City, Iowa, United States of America, 2 Department of Neurosurgery, Stanford University School of Medicine, Stanford, California, United States of America

* andy-frank@uiowa.edu

## Abstract

Neurons require high amounts of energy, and mitochondria help to fulfill this requirement. Dysfunctional mitochondria trigger problems in various neuronal tasks. Using the *Drosophila* neuromuscular junction (NMJ) as a model synapse, we previously reported that Mitochondrial Complex I (MCI) subunits were required for maintaining NMJ function and growth. Here, we report tissue-specific adaptations at the NMJ when MCI is depleted. In *Drosophila* motor neurons, MCI depletion causes profound cytological defects and increased mitochondrial reactive oxygen species (ROS). But instead of diminishing synapse function, high levels of neuronal mitochondrial ROS trigger a homeostatic signaling process that maintains normal NMJ excitation. We identify molecules mediating this compensatory response. MCI depletion in muscles also enhances local mitochondrial ROS. But high levels of muscle mitochondrial ROS cause destructive responses: synapse degeneration, mitochondrial fragmentation, and impaired neurotransmission. In humans, mutations affecting MCI subunits cause severe neurological and neuromuscular diseases. The tissue-level effects that we describe in the *Drosophila* system are potentially relevant to forms of mitochondrial pathogenesis.

## Introduction

Neurons have vast energy needs. These needs are primarily satisfied by healthy pools of mitochondria [1,2]. Mitochondria generate energy through the action of the ATP synthase complex in the electron transport chain [3,4]. They also perform complementary functions, including maintaining calcium homeostasis [5,6], promoting cell survival [7], triggering reactive oxygen species (ROS) signaling [8], stimulating lipid synthesis [9], and regulating innate immunity [10]. For energy-driven neurons, it is thought that the primary role of mitochondria is to provide ATP. It is less understood how other mitochondrial functions contribute to the regulation of normal

**Data availability statement:** All relevant data are within the paper and its Supporting information files. Quantified data are plotted and displayed in the main and supplemental figures. Summary data of the figures are displayed in Tables A–U in S1 Tables. Reagents and sources for generating the data are shown in Table V in S1 Tables. Raw data for all figures are in supplemental Excel files. The S1 Data Excel file contains raw data for the main figures. The S2 Data Excel file contains raw data for the supplemental figures.

**Funding:** B.M. was supported in part by NIH/NINDS (https://www.ninds.nih.gov) grants to C.A.F. (NS085164, NS130108, and NS136753). Collectively, the work was supported by those same NIH/NINDS grants, as well as funds from the University of Iowa's Carver College of Medicine (https://medicine.uiowa.edu) to C.A.F. Funding supporting the work executed by S.A.B. and X.W. was from an NIH/NINDS (https://www.ninds.nih.gov) R01 grant NS128040 to X.W. The funders had no role in study design, data collection and analysis, decision to publish, or preparation of the manuscript.

**Competing interests:** The authors have declared that no competing interests exist.

**Abbreviations:** BL, Bloomington Line; BDSC, Bloomington Drosophila Stock Center; BMP, bone morphogenetic protein; CSP, cysteine string protein; EPSP, excitatory postsynaptic potentials; ER, endoplasmic reticulum; $IP_3R$, $IP_3$ receptor; LTP, long-term potentiation; MAGUK, membrane-associated guanylate kinase; MCU, mitochondrial calcium uniporter; MCI, mitochondrial complex I; mEPSP, miniature excitatory postsynaptic potentials; Mfn1, mitofusin 1; Mfn2, mitofusin 2; NACA, N-acetyl cysteine amide; NMJ, neuromuscular junction; OCR, oxygen consumption rate; PHP, presynaptic homeostatic potentiation; QC, quantal content; RET, reverse electron transfer; ROS, reactive oxygen species; RRP, readily releasable pool; RyR, ryanodine receptor; SSR, subsynaptic reticulum; STAR, structured, transparent, accessible reporting; TOR, target of rapamycin; UPR, unfolded protein response; VNC, ventral nerve cord.

neurophysiology. It is also not well understood how neural tissues or synaptic sites cope when they are challenged with a loss of mitochondria. Genetic models can help to address these puzzles.

Mitochondrial Complex I (MCI) (NADH ubiquinone oxidoreductase) is an essential part of the electron transport chain and ATP production, and it comprises dozens of distinct subunits. Much of our understanding about MCI derives from systemic analyses of its assembly. Studies have been performed on Complex I components from diverse organisms, including *Neurospora crassa* and *Drosophila melanogaster*. Those studies demonstrate that discrete MCI subunits are ancient; indeed, there are few differences between these MCI models from simple organisms and the corresponding human and bovine orthologs [11–14]. For *Drosophila melanogaster*, MCI consists of at least 42 distinct subunits; the 14 core MCI subunits are present, as are at least 28 accessory subunits [12].

In humans, MCI dysfunction has been linked to diseases such as Leigh syndrome, mitochondrial myopathy, and encephalomyopathy, as well as forms of stroke [15–18]. On a cellular level, MCI dysfunction can cause the demise of neurons and muscles; these phenotypes are typically attributed to defects in ATP production [19,20]. However, in addition to the ATP production defects, mutations affecting MCI subunit components are also associated with excess mitochondrial ROS. Normally, ROS accumulation can be neutralized by the cellular antioxidant system [21]. But if that system becomes overwhelmed, there can be consequences for cells and organ systems—including progressive neurodegeneration and seizures for the nervous system [22–24]. On the level of synapses, it is possible that MCI loss triggers severe molecular consequences, and it is also possible that excess ROS plays a role.

For a previous paper, we depleted MCI function at the *Drosophila* neuromuscular junction (NMJ). Our data suggested fundamental synaptic functions for MCI [13]. Here we expand upon that work, mostly taking advantage of RNAi-mediated depletion of the *Drosophila* homologs of human NDUFS7. We also scrutinize loss-of-function mutants of other MCI subunits and pharmacological inhibition of MCI. Our collective data show that MCI depletion causes *Drosophila* phenotypes reminiscent of mitochondrial diseases, such as progressive degeneration of muscle and presynaptic cytoskeleton, excess ROS production, loss of mitochondria, and alteration in mitochondrial morphology.

On single-tissue levels, we were surprised to find that there were opposite effects on synapse activity in the presynaptic motor neurons versus the postsynaptic muscles. MCI dysfunction in *Drosophila* motor neurons causes profound cytological phenotypes, but there are no significant functional phenotypes. This appears to be because neuronal mitochondrial ROS triggers an adaptive response, demonstrated visually by active zone enhancement. This ROS-driven enhancement of active zones occurs through at least two processes: 1) regulation of calcium flux from intracellular stores (ER) and mitochondria and 2) use of glycolysis as an alternative energy source. By contrast, postsynaptic depletion of MCI and the associated elevation of muscle ROS trigger a destructive response: disruption of NMJ morphology and the Dlg-Spectrin scaffold that is critical for normal active zone-receptor apposition.

To our knowledge, these cellular and molecular mechanisms of MCI deficiency have not previously been elucidated at synapse-specific or tissue-specific levels.

## Results

### One RNAi transgene targets two *Drosophila* homologs of a core MCI subunit, NDUFS7

We previously reported impairments in NMJ synapse development and function when Mitochondrial Complex I (MCI) was depleted [13]. We observed robust synaptic phenotypes when driving a transgenic RNAi line against an MCI subunit, termed *UAS-ND-20L[RNAi]* (*ND-20L*<sup>HMJ23777</sup>) [13]. That RNAi line encodes a short hairpin that matches 21 consecutive nucleotides of the *ND-20L* gene. Recent work indicates that *ND-20L* is only sparsely expressed in *Drosophila* tissues [25–27]. However, the same short hairpin also matches 19 consecutive nucleotides of the *ND-20* gene, and *ND-20* is ubiquitously expressed [25–27]. *ND-20* and *ND-20L* encode *Drosophila* homologs of the core NDUFS7 MCI subunit. Because both genes are effectively "on-target" for the RNAi transgene, we term the *UAS-ND-20L[RNAi]* transgene as *UAS-NDUFS7[RNAi]* for this study.

We used the GAL4/UAS system to verify that *UAS-NDUFS7[RNAi]* can diminish MCI function in relevant tissues. We drove its expression using neuronal or muscle GAL4 drivers. By quantitative RT-PCR, message levels of both NDUFS7-encoding genes were significantly diminished in those tissues in larvae (S1A–S1D Fig). *ND-20L* message levels were down in both neurons and muscle, but variable in neurons, likely due to the low endogenous expression of *ND-20L* (S1A and S1B Fig). *ND-20* message levels were significantly knocked down by the transgene in both tissues (S1C and S1D Fig). We also tested if the transgene could diminish mitochondrial respiration by a Seahorse assay. To acquire sufficient material for the assay, we collected mitochondria from adult muscles (see STAR methods). Driving the *UAS-NDUFS7[RNAi]* transgene in adult muscle starkly reduced oxygen consumption rate (OCR) in muscle mitochondria across all time points tested (S1E and S1F Fig). These assays confirmed that *UAS-NDUFS7[RNAi]* targets MCI.

### Depletion of MCI affects mitochondrial integrity in multiple *Drosophila* synaptic tissues

We wanted to understand what was happening to mitochondria to affect synapse function when MCI was depleted. We started by examining mitochondria by microscopy. To visualize them, we expressed a *UAS-Mito-GFP* transgene [28] in *Drosophila* tissues. Concurrently, we used tissue-specific GAL4 drivers alone (as controls) or GAL4 drivers + *UAS-NDUFS7[RNAi]* to deplete MCI function [13]. With these tools, we made qualitative observations of mitochondrial morphology (Fig 1), and we quantified those observations in subsequent analyses.

In motor neurons, the Mito-GFP signal localized to the neuropil of the VNC (Fig 1A–1A′ and 1B–1B′). In control neurons, the neuropil mitochondria had a filamentous appearance. By contrast, *NDUFS7*-depleted neurons had punctate and clustered mitochondria. (Fig 1E–1E′ and 1F–1F′). We examined mitochondria in the motor axons that innervate proximal and distal NMJs (Fig 1C–1C′, 1D–1D′, 1G–1G′, and 1H–1H′). The proximal segment A2 axons had abundant mitochondria in all cases (Fig 1C–1C′ and 1G–1G′). However, for the distal segment A5 axons, *NDUFS7* depletion elicited an obvious decrease in mitochondria number (Fig 1D–1D′ and 1H–1H′). This A2 versus A5 discrepancy was consistent with prior work by others examining defects in mitochondrial trafficking dynamics: distal sites can show phenotypes more prominently [29,30].

We hypothesized that fewer mitochondria in the A5 axon might correlate with a neurotransmission defect at the NMJ. Yet by NMJ electrophysiology, we found no significant differences in the evoked amplitude compared to the control NMJs in the distal segment A5 (Fig 1I–1K). These data matched our prior examination of MCI at the A2 and A3 segments of the NMJ, where neuronal impairment of MCI was not sufficient on its own to reduce evoked NMJ neurotransmission [13].

In muscle, we observed an array of phenotypes. As with neurons, there were clustered mitochondria when *NDUFS7* gene function was depleted (Fig 1L and 1M). Additionally, there was a tissue-level phenotype: *NDUFS7*-depleted muscles

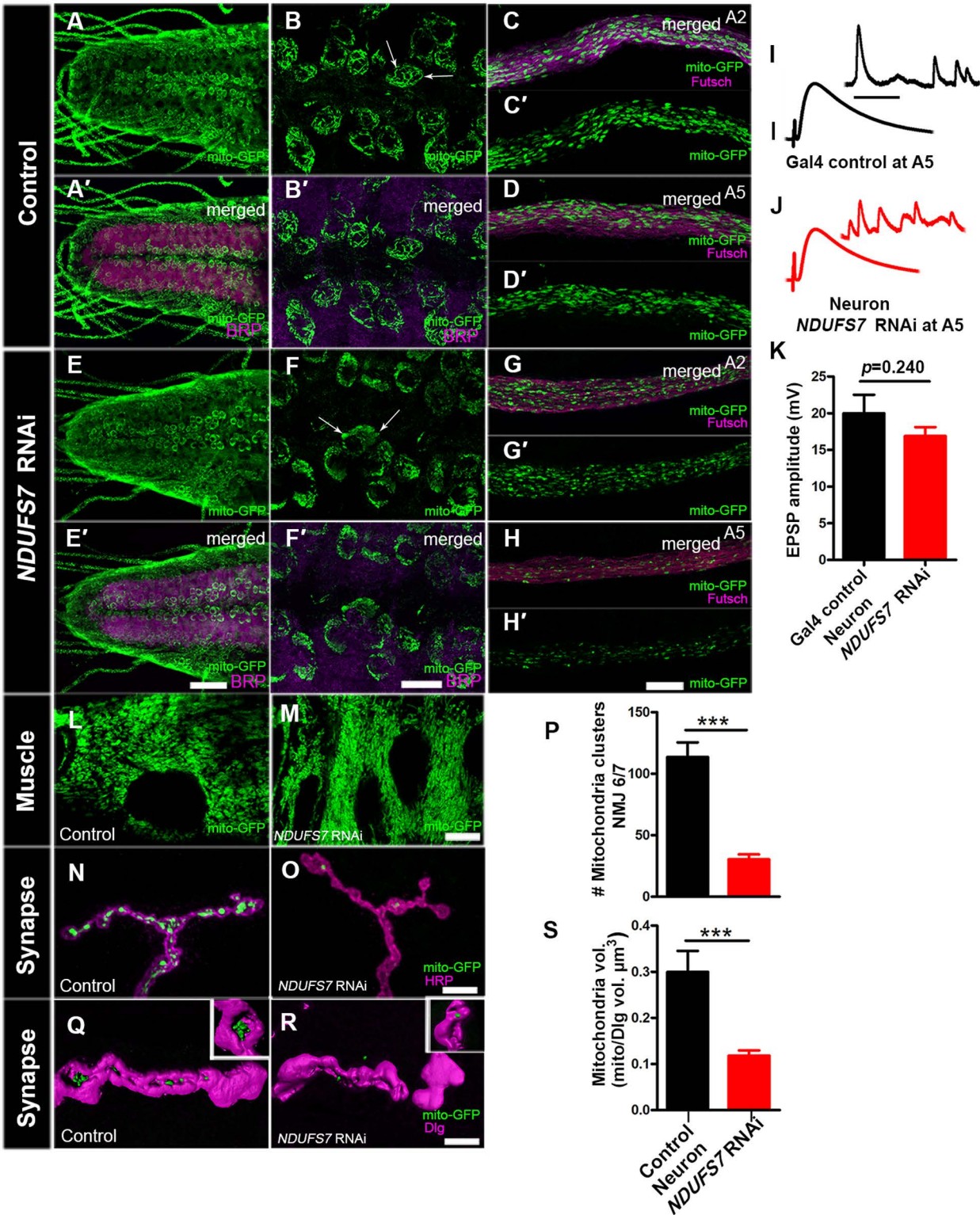

**Fig 1. MCI-depleted flies harbor fewer mitochondria at neuromuscular junctions.** Mitochondrial morphology and trafficking defects in the ventral nerve cord (VNC), distal axons, and boutons. RNAi lines and controls were crossed to a motor neuron driver (*D42-GAL4*) and a mitochondrial marker (*UAS-mito-GFP*). (**A, A′**) control VNC, (**B, B′**) a magnified section of VNC, (**C, C′**) proximal (A2) axons, and (**D, D′**) distal (A5) axons. These tissues

exhibit regular mitochondrial clusters in the soma and axons. **(E, E′)** *NDUFS7* knockdown in the VNC, **(F, F′)** a magnified section of VNC, **(G, G′)** proximal (A2) axons, and **(H, H′)** distal (A5) axons. Mitochondria are abnormally clustered from *NDUFS7[RNAi]* expression in the VNC and distal segments of A5 axons. *NDUFS7[RNAi]* expression also yields fewer mitochondria in the distal segments when compared to the proximal segments. **(I, J)** Representative electrophysiological traces showing evoked potentials of *mitoGFP, D42-Gal4 × UAS-NDUFS7[RNAi]* larvae at the A5 hemisegment of muscle 6/7 synapse. Scale bars for EPSPs (mEPSP) are $x = 50$ ms (1,000 ms) and $y = 10$ mV (1 mV). Fewer mitochondria at the presynaptic A5 hemisegment did not affect evoked NMJ excitation. **(K)** Quantification showing EPSP amplitude at NMJ 6/7 in control (*D42-Gal4*/+; EPSP: 19.99 mV ± 2.53, $n = 6$) and RNAi-depleted animals (*NDUFS7[RNAi]*/+; *D42-Gal4*/+;EPSP: 16.88 mV ± 1.21, $n = 9$). **(L, M)** Representative images showing mitochondrial morphology in control and RNAi-depleted animals in muscles. Mitochondria in *NDUFS7-depleted* larval muscles are clustered compared to the control muscles. **(N–Q)** *NDUFS7[RNAi]* expression yields almost no mitochondria in boutons when co-stained with pre- (HRP) or postsynaptic markers (Discs Large, Dlg). (A–H′, L–M, and N–O) Scale bar: 10 µm. **(P)** Quantification showing the number of mitochondrial clusters at NMJ 6/7 in control (*D42-Gal4*/+; # clusters: 113.6 ± 11.97, $n = 7$) and RNAi-depleted animals (*NDUFS7[RNAi]*/+; *D42-Gal4*/+; # clusters: 30.33 ± 4.02, $n = 6$). **(Q, R)** 3D-rendered image revealing the volume (µm 3) of mitochondria at NMJ 6/7 in RNAi knockdown larvae (*NDUFS7[RNAi]*/+; *D42-Gal4*/+; 0.11 ± 0.01 µm 3, $n = 16$) compared to the driver control animals (*D42-Gal4*/+; 0.29 ± 0.04 µm 3, $n = 14$). Insets are large boutons from the same images rotated to display 3D-rendered mitochondrial clusters at these synaptic terminals. Scale bar: 5 µm. **(S)** Quantification shows a significantly lower mitochondria volume in boutons at NMJ 6/7 in *NDUFS7-depleted* neurons. ***$p < 0.0001$ and ***$p = 0.0003$ for mitochondrial clusters and volume, respectively. Statistical analyses based on Student $t$ test. Error bars represent mean ± s.e.m. Raw data for this figure are available in the S1 Data Excel file, tab Fig 1.

were developed, but they looked disorganized and fragmented, with oblong-shaped nuclei in the muscle syncytia (Fig 1L and 1M). This phenotype could explain why we previously observed that muscle impairment of MCI was sufficient to reduce evoked NMJ neurotransmission [13].

To examine the mitochondria at presynaptic NMJ release sites, we used the motor neuron GAL4 driver to label NMJ boutons with Mito-GFP. For image analysis, we marked the presynaptic membrane boutons with anti-HRP immunostaining. Control NMJs contained abundant and large clusters of mitochondria in synaptic boutons, but by comparison, *NDUFS7[RNAi]* boutons contained small clusters and few mitochondria (Fig 1N and 1P). We measured the mitochondrial volume in a three-dimensional stack and compared it to the synaptic volume (Fig 1Q–1S). The Mito-GFP signal occupied a sizeable proportion of the bouton volume in controls (~30%), but this value was significantly diminished in *NDUFS7*-depleted animals (~10%) (Fig 1S). Collectively, our data suggest that the depletion of *NDUFS7* by RNAi leads to abnormal mitochondrial clustering in the neuronal cell body and muscle—as well as losses of distal axon and synaptic mitochondria.

## Loss of MCI phenocopies loss of Mitofusin

The cell-level NDUFS7-depletion phenotypes were reminiscent of *Drosophila* mutants impairing mitochondrial dynamics [29,31]. Therefore, we re-examined MCI-depleted mitochondria, this time additionally impairing genes known to mediate mitochondrial fusion and fission. Mitofusin 1 (Mfn1) and Mitofusin 2 (Mfn2) are GTPases that regulate outer mitochondrial membrane fusion [32,33]. The *Drosophila* gene encoding the Mitofusin homolog is called *marf*. Dynamin-related protein 1 is a GTPase that regulates mitochondrial fission. In *Drosophila*, this factor is encoded by the gene *drp1* [34]. Previous work reported that defective fusion results in fragmented mitochondria, while defective fission can lead to enlarged mitochondria [35]. We used RNAi-mediated knockdown constructs for each of these genes.

As before, we observed that wild-type motor neurons had filamentous and oval mitochondria, while *NDUFS7*-depleted neurons had fewer and smaller clustered mitochondria in the ventral nerve cord (VNC) (Fig 2A) and axons (Fig 2B). Knockdown of the fusion gene *marf* phenocopied *NDUFS7* loss, revealing small mitochondria in motor neurons, while knockdown of the fission gene *drp1* yielded filamentous mitochondria (Fig 2A). Simultaneously depleting motor neurons of *marf* and *NDUFS7* by RNAi did not show any additive defect in mitochondrial appearance in the VNC and axons (Fig 2A and 2B). This result could mean that the genes share a common process to regulate mitochondrial fusion. By contrast, depleting *drp1* and *NDUFS7* by RNAi simultaneously yielded punctate mitochondria. This result likely means that the punctate *NDUFS7* mitochondrial phenotypes (potential fusion phenotypes) are epistatic to *drp1* loss (Fig 2A and 2B).

We measured mitochondrial branch length from skeletonized images of the mitochondria (Skeletonize3D, ImageJ plugin, data in Tables A, B, and C in S1 Tables). Control and *drp1* knockdown showed normal mitochondrial branch length,

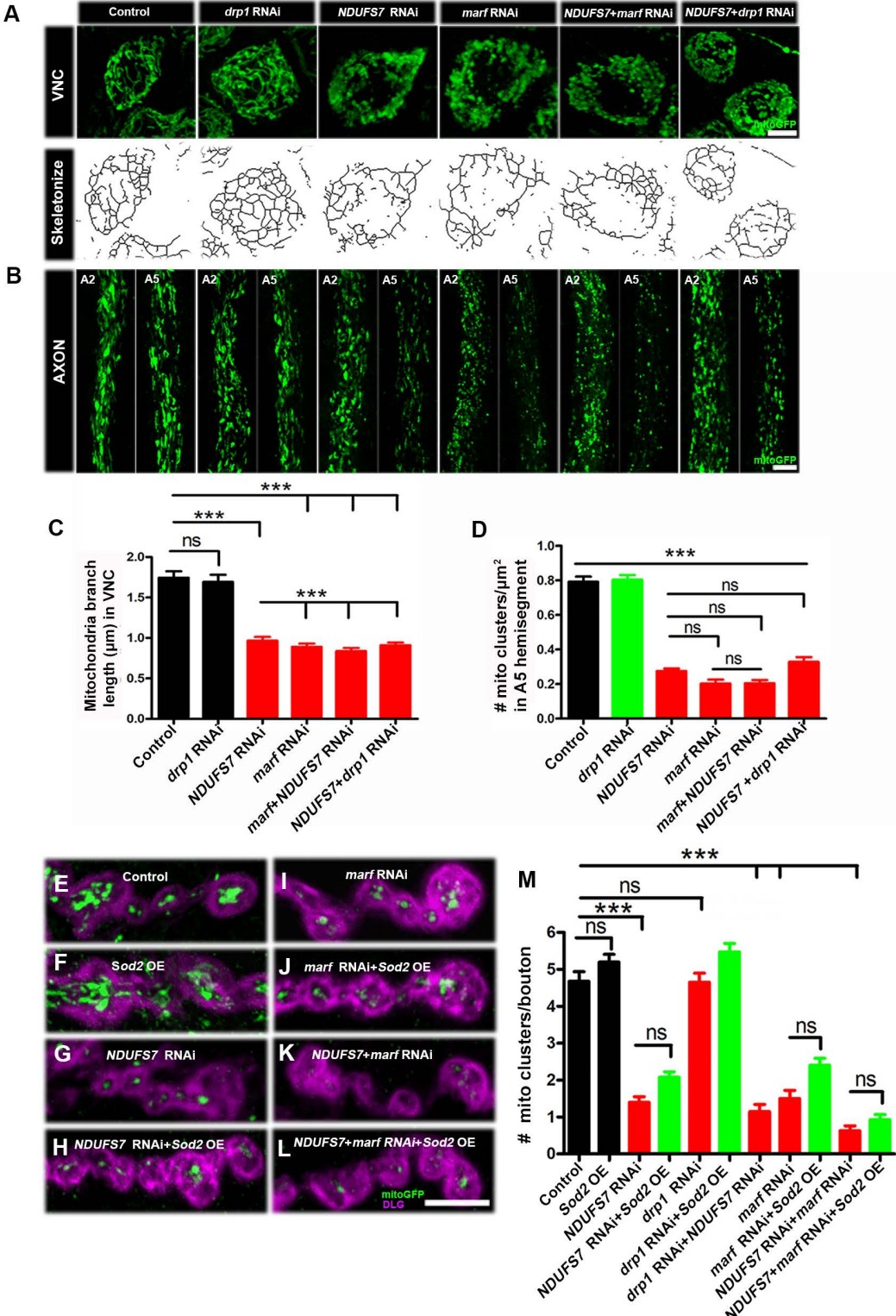

**Fig 2. Loss of *NDUFS7* in motor neurons phenocopies a *marf* depletion.** Mitochondrial morphology and trafficking defects in the ventral nerve cord and distal axons. To label neuronal mitochondria, *UAS-[RNAi]* lines and controls were crossed to a motor neuron driver (*D42-Gal4*) and a mitochondrial marker (*UAS-mitoGFP*). **(A)** Ventral nerve cord (VNC): *UAS-mitoGFP* and *drp1[RNAi]* exhibit normal mitochondrial organization, *NDUFS7[RNAi]* and

*marf*[RNAi] exhibit clustered mitochondria, *NDUFS7[RNAi]; marf[RNAi]* and *NDUFS7[RNAi]; drp1[RNAi]* doubles exhibit clustered mitochondria in the soma. The fluorescent images were subsequently skeletonized to measure mitochondrial branch length. **(B)** Comparison of a proximal axonal segment in A2 and a distal segment in A5. Distal segments of A5 axons in *NDUFS7[RNAi]* and *marf[RNAi]* contain many fewer mitochondria than proximal segments. Knocking down *NDUFS7[RNAi]* and *marf[RNAi]* together does not show an additive effect. (A, B) Scale bar: 10 μm. **(C, D)** Histogram showing mitochondrial branch length (μm) and number (μm 2 area of bouton) in VNC and axons of the third instar larvae in the indicated genotypes. **(E–L)** Representative images of the A2 hemisegment of muscle 6/7 NMJs in (E) *UAS-mito-GFP, D42-Gal4/+*, (F) *UAS-Sod2/+; UAS-mito-GFP, D42-Gal4/+*, (G) *NDUFS7[RNAi]/+; UAS-mito-GFP, D42-Gal4/+*, (H) *NDUFS7[RNAi]/UAS-Sod2; UAS-mito-GFP, D42-Gal4/+*, (I) *UAS-mito-GFP, D42-Gal4/marf[RNAi]*, (J) *UAS-Sod2/+; UAS-mito-GFP, D42-Gal4/marf[RNAi]*, (K) *NDUFS7[RNAi]/+; UAS-mito-GFP, D42-Gal4/marf[RNAi]*, and (L) *NDUFS7[RNAi]/UAS-Sod2; UAS-mito-GFP, D42-Gal4/marf[RNAi]* larvae immunostained with antibodies against HRP (magenta) and GFP (mito-GFP:green) to label neurons and mitochondria. *NDUFS7[RNAi]*- and *marf[RNAi]*-depleted animals harbor fewer mitochondria at the terminals as compared to control animals. Transgenic *UAS-Sod2* did not rescue mitochondrial clustering defects. (E–L) Scale bar: 5 μm. **(M)** Histograms showing quantification of mitochondrial clusters at the NMJs in the indicated genotypes. ***$p < 0.0001$; ns, not significant. Statistical analysis based on one-way ANOVA followed by post-hoc Tukey's multiple-comparison test. Error bars represent mean ± s.e.m. Raw data for this figure are available in the S1 Data Excel file, tab Fig 2.

but knockdown using *NDUFS7[RNAi]* or *marf[RNAi]*—or knockdowns using combinations of each—exhibited short branch length (Fig 2C and Table A in S1 Tables).

To quantify mitochondria in axons, we counted Mito-GFP-positive puncta in distal A5 motor axons labeled by anti-GFP. Control axons and *drp1*-depleted axons contained abundant mitochondria (Fig 2D and Table A in S1 Tables). By contrast, any gene manipulation or combination targeting *NDUFS7* or *marf* by RNAi resulted in diminished numbers of mitochondrial clusters (Fig 2D and Table A in S1 Tables).

We extended the analysis to NMJ terminals. We counted Mito-GFP clusters in presynaptic boutons apposed by postsynaptic densities, labeled by anti-Discs Large 1 (Dlg1) (Fig 2E–2M and Table D in S1 Tables). The results matched our earlier observations (Fig 1N–1S). Control NMJs and *drp1*-depleted NMJs contained numerous mitochondrial clusters per bouton (Fig 2E and 2M). However, *NDUFS7*-depleted boutons contained few mitochondria, and this was phenocopied by *marf[RNAi]* (Fig 2E–2M and Table D in S1 Tables). Collectively, these results suggest that *Drosophila* NDUFS7 (and hence MCI) contributes to normal mitochondrial fusion, likely in conjunction with the Mitofusin homolog, Marf.

## Mitochondrial ROS contribute to synaptic phenotypes

Several studies have demonstrated that Complex I loss results in high levels of mitochondrial ROS [36–40]. This means that excess ROS could be contributing to the cytological and mitochondrial fusion phenotypes that we have described.

We checked if we could observe mitochondrial ROS (superoxide) in living *Drosophila* tissue and if ROS levels corresponded to Complex I function (S1 Fig). We used a commercially available fluorescent mitochondrial superoxide indicator, MitoSOX (ThermoFisher, STAR methods) [41–43]. With MitoSOX, we observed mitochondrial superoxide in many tissues by fluorescence microscopy. There was a baseline level of ROS in controls (S2A, S2E, S2I, and S2J Fig and Table E in S1 Tables), and the level was greatly increased in *NDUFS7*-deficient motor neuron cell bodies and muscle (S2B, S2F, S2I, and S2J Fig and Table E in S1 Tables).

Next, we tested if ROS scavengers could reverse the high MitoSOX fluorescence levels in *NDUFS7*-deficient tissues. We fed a pharmacological scavenger, N-Acetyl Cysteine Amide (NACA) [44–46] to *Drosophila* larvae (STAR methods). We also used a transgene, *UAS-Sod2* [47], to express a superoxide dismutase enzyme. Both successfully diminished the high levels of mitochondrial ROS that resulted from *NDUFS7* depletion at the NMJ, and both worked in muscle and neurons (S2C, S2D, and S2G–S2J and Table E in S1 Tables). We also tested a related idea: if genetic *Sod2* knockdown by RNAi could phenocopy loss of MCI by enhancing MitoSOX levels in these same tissues. It did: Transgenic *UAS-Sod2 [RNAi]* enhanced MitoSOX signal in all tissues and subcompartments, including the VNC (S3A–S3C Fig and Table F in S1 Tables), the motor neuron axons (S3D–S3F Fig and Table F in S1 Tables), the synaptic boutons (S3G–S3I Fig and Table F in S1 Tables), and the muscle (S3J–S3L Fig and Table F in S1 Tables).

Next, we tested if these same ROS scavengers could reverse mitochondrial phenotypes caused by loss of MCI. Co-expressing *UAS-Sod2* or rearing larvae with NACA suppressed the mitochondrial morphology defects in the VNC of *NDUFS7*-depleted animals; it also restored axonal loss of mitochondria (S4A–S4D Fig and Table B in S1 Tables). To test an additional MCI manipulation, we knocked down *Drosophila ND-30* (homologous to human *NDUFS3*) in motor neurons. As with *NDUFS7*, depleting *ND-30* in motor neurons yielded punctate mitochondria in the VNC, but the addition of *UAS-Sod2* restored a wild-type, filamentous mitochondrial morphology (S4A and S4C Fig). Similarly, loss of *ND-30* gene function in neurons depleted A5 axonal mitochondria and this phenotype was also reversed by *UAS-Sod2* transgenic expression (S4B and S4D Fig and Table B in S1 Tables). In contrast to *UAS-Sod2*, neither *UAS-Sod1* nor *UAS-Catalase* worked to reverse mitochondrial morphology defects (S5 Fig and Table C in S1 Tables). SOD1 (cytosol) and Catalase (peroxisome) localize to different compartments than SOD2 (mitochondrial matrix). Therefore, these results could indicate that a scavenger needs to access the proper compartment for rescue.

Because of the links between MCI and mitochondrial fusion, we considered whether a *marf* loss of function could also yield high levels of neuronal ROS (S6 Fig). It did: both *marf* and *NDUFS7* loss-of-function conditions showed high levels of mitochondrial superoxide in motor neuron cell bodies (S6A–S6D, S6A′–S6D′ and S6M Fig and Table E in S1 Tables) in motor axons (S6E–S6H, S6E′–S6H′ and S6N Fig and Table E in S1 Tables); and at NMJ sites (S6I–S6L, S6I′–S6L′, and S6O Fig and Table E in S1 Tables). These ROS phenotypes were not confined to genetic manipulations. We made similar observations when MCI was impaired pharmacologically by feeding rotenone to developing larvae (S6 Fig and Table E in S1 Tables). In the case of rotenone, the amount of mitochondrial ROS in the tissues was high, but it was generally not increased as much as with the genetic manipulations (S6M–S6O Fig and Table E in S1 Tables).

Finally, because the distal A5 motor axons accumulated high levels of ROS when subjected to these insults, we examined them for synaptic vesicle trafficking defects. We immunostained for Cysteine String Protein (CSP), a DNAJ-like co-chaperone and synaptic vesicle-associated protein (CSP). *NDUFS7* depletion caused aberrant accumulation of CSP in the A5 motor axons; and this defect was suppressed by motor neuron transgene overexpression of *UAS-Sod2* or by feeding animals with NACA (S7A–S7L Fig and Table G in S1 Tables). However, this phenotype was not suppressed by motor neuron overexpression of *UAS-Sod1* or *UAS-Catalase* (S8 Fig and Table H in S1 Tables).

ROS scavengers did not reverse all mitochondrial abnormalities. For example, expressing *UAS-Sod2* in the *UAS-NDUFS7[RNAi]* or *UAS-marf[RNAi]*-depletion backgrounds did not restore mitochondrial clusters to motor neuron terminals (Fig 2E–2M and Table D in S1 Tables). For the remainder of the study, we used scavengers as complementary tools to test which MCI-loss phenotypes were likely due to mitochondrial ROS.

## Loss of MCI subunits impairs synaptic cytoskeletal stability

ROS can modulate the cytoskeleton, either through redox modification of cytoskeletal proteins or by altering pathways that regulate cytoskeletal organization [48]. To test whether the mitochondrial defects and abnormal accumulation of ROS were associated with the altered synaptic cytoskeleton, we labeled synaptic boutons with an anti-Futsch antibody (Fig 3). Futsch is a *Drosophila* MAP1B homolog that associates with microtubules [49].

In motor neuron Gal4-control and *UAS-Sod2* overexpression larvae, Futsch organized in periodic loops, as expected from previous characterizations [49] (Fig 3A and 3B and Table G in S1 Tables). But in NDUFS7-depleted larvae, the anti-Futsch staining showed a significant reduction in microtubule loops (Fig 3C and Table G in S1 Tables). These *NDUFS7* phenotypes were suppressed by motor neuron overexpression of *UAS-Sod2* or by raising animals on food containing 0.5 mM NACA (Fig 3D and 3E and Table G in S1 Tables). However, they were not significantly restored by neuronal overexpression of *UAS-Sod1* or *UAS-Catalase* (S9 Fig and Table H in S1 Tables). These data indicate that loss of MCI regulates cytoskeletal architecture due to excessive accumulation of mitochondrial ROS in neurons.

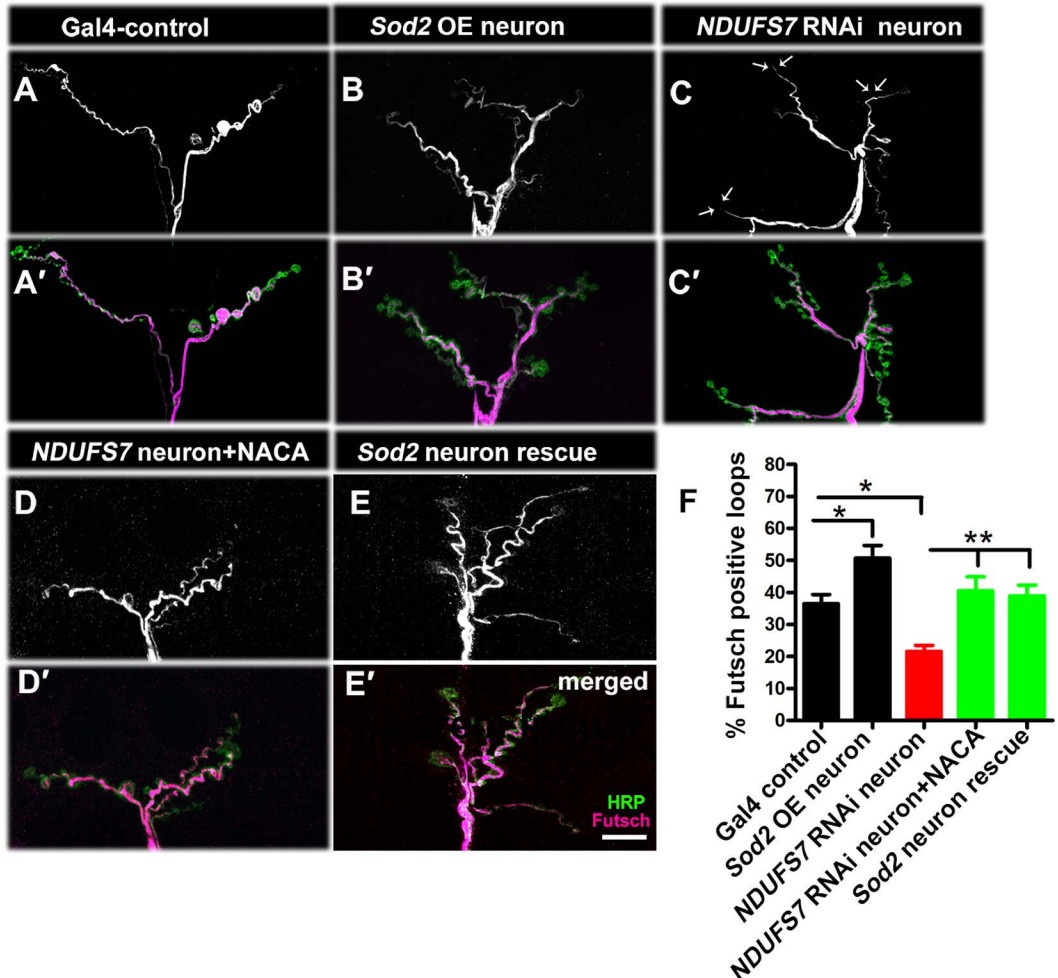

**Fig 3. *NDUFS7* depletion in motor neurons affects synapse stability.** Representative confocal images of NMJ synapses at muscle 6/7 of (**A, A′**) *D42-Gal4* control, (**B, B′**) *UAS-Sod2* overexpression (**C, C′**) *D42-Gal4*-driven *NDUFS7[RNAi]* (*NDUFS7[RNAi]/+; D42-Gal4/+*), (**D, D′**) *NDUFS7* knock-down with NACA rescue (*NDUFS7[RNAi]/+; D42-Gal4/+* with NACA), (**E, E′**) *NDUFS7* knockdown with *UAS-Sod2* (*UAS-NDUFS7[RNAi]/UAS-Sod2; D42-Gal4/+*). Each condition was double immunolabeled with 22C10 (anti-Futsch, magenta) and anti-HRP (green) antibodies. The motor neuron-depleted *NDUFS7[RNAi]* larvae showed a decrease in the number of Futsch-positive loops as compared to the Gal4 control. Futsch-positive loops were significantly restored to the control number when *NDUFS7[RNAi]* knockdown larvae were raised in media containing NACA or when genetically expressing *UAS-Sod2* in the *UAS-NDUFS7[RNAi]* knockdown background. Scale bar: 10 μm. **(F)** Histograms showing the percentage of Futsch-positive loops in the indicated genotypes. *$p = 0.01$ (Gal4 control vs. *Sod2* OE neuron), *$p = 0.0008$ (Gal4 control vs. *NDUFS7[RNAi]* neuron), **$p = 0.001$ (*NDUFS7[RNAi]* neuron vs. *NDUFS7[RNAi]* neuron with NACA) and **$p = 0.0005$ (*NDUFS7[RNAi]* neuron vs. *Sod2* neuron rescue). Statistical analysis based on one-way ANOVA followed by post-hoc Tukey's multiple-comparison test. Error bars represent mean ± s.e.m. Raw data for this figure are available in the S1 Data Excel file, tab Fig 3.

## Modest neurotransmission phenotypes after motor neuron-specific loss of MCI or Marf

Given the cytological phenotypes after neuronal MCI loss, it was puzzling that there seemed to be little-to-no electrophysiological consequence at NMJs ([13] and Fig 1J). We probed this finding, this time depleting motor neurons of *marf* and/or *NDUFS7* gene function. We recorded spontaneous miniature excitatory postsynaptic potentials (mEPSP) and excitatory postsynaptic potentials (EPSP).

Phenotypes were normal-to-mild (S10 Fig). For both *NDUFS7[RNAi]* and *marf[RNAi]*, there were small, but statistically significant decreases in mEPSP amplitude (S10A–S10C and S10G Fig and Table I in S1 Tables). But for *NDUFS7[RNAi]*, evoked events (EPSP) and calculated quantal content (QC) were at control levels (S10A, S10B, S10H, and S10I Fig and Table I in S1 Tables). For neuronal *marf[RNAi]*, those measures were near-normal, with a slight decrease in EPSP amplitude (S10C and S10H Fig and Table I in S1 Tables) and a slight increase in QC (S10I Fig and Table I in S1 Tables). Finally, for a double *NDUFS7[RNAi]* + *marf[RNAi]* knockdown condition in neurons, we observed a small, but statistically significant decrease in EPSP amplitude (S10E and S10H Fig and Table I in S1 Tables).

Synaptic phenotypes caused by mitochondrial dysfunction might be masked until synapses are challenged with extreme conditions, like high-frequency stimulation [50,51]. Therefore, we challenged neuronally *NDUFS7*-depleted *Drosophila* NMJs in several ways (S11 Fig). First, we lowered recording saline [$Ca^{2+}$] to 0.15 mM, which is roughly one order of magnitude lower than physiological calcium. In low calcium, the motor neuron-driven *NDUFS7[RNAi]* NMJs had slightly smaller evoked potentials compared to driver controls, but the numerical reduction was not statistically significant (S11A–S11C Fig and Table J in S1 Tables). Next, we lowered extracellular [$Ca^{2+}$] even further, to 0.1 mM, which yielded a mix of successful EPSP firing events and failures. Failure analyses revealed a decrease in failure rate at the neuronally depleted *NDUFS7[RNAi]* NMJs compared to control animals (S11D Fig and Table J in S1 Tables). This was slightly surprising because it potentially demonstrates an increased probability of release. Higher failure rates in 0.1 mM calcium were restored when *NDUFS7*-depleted larvae were raised in a media containing NACA or genetically expressing *Sod2* in motor neurons (S11D Fig and Table J in S1 Tables), indicating that presynaptic release probability could be related to mitochondrial ROS levels.

Finally, we checked if forms of short-term neuroplasticity were affected by *NDUFS7* loss in motor neurons. For two different extracellular [$Ca^{2+}$] conditions (0.4–1.5 mM), we did not observe any significant changes in paired-pulse ratios (S11E–S11J Fig and Table J in S1 Tables). Likewise, we did not note depreciation of evoked neurotransmission over the course of high-frequency stimulus trains in high calcium (S11K–S11P Fig). Collectively, these data suggest that there might be small effects on NMJ physiology due to loss of MCI or defective mitochondrial fusion and ROS production. But the aggregate data also indicate that neuronal mitochondrial defects alone do not drastically affect NMJ neurotransmission.

## Loss of MCI in neurons controls the level and distribution of the active zone to stabilize synaptic strength

We wondered how the NMJ synapse might evade severe dysfunction—and even show resilience during a failure analysis—despite loss of mitochondria in the motor neurons and synaptic terminals. One possibility is that *NDUFS7* loss/MCI impairment could trigger a form of functional homeostatic compensation of the NMJ. Another idea is that the mitochondrial ATP generated is superfluous at the NMJ—and that any energy-intensive functions that mitochondria support could be redundantly covered by glycolysis. These models are not mutually exclusive, and for any scenario, mitochondrial ROS downstream of defective MCI could be a candidate signal. Recent findings have demonstrated that ROS intermediates, mitochondrial distribution, and mitochondrial trafficking all affect development of the *Drosophila* NMJ [48,52,53].

To probe this idea, we imaged the presynaptic active zone apparatus in neuronally depleted *NDUFS7[RNAi]* flies. Third-instar larval active zones showed a decrease in active zone protein Bruchpilot (BRP) puncta density per unit area in *NDUFS7*-depleted NMJs compared to control NMJs (Fig 4A–4C and 4G–4I and Table K in S1 Tables). But they also showed a robust enhancement phenotype: a 40% increase in active zone (BRP) immunofluorescence signal per unit area, compared to control by laser scanning confocal microscopy (Fig 4A–4C, 4F, 4H, and 4I and Table K in S1 Tables). This result was intriguing because NMJ active zone enhancements (or changes in active zone sub-structure) have been proposed by other labs to be molecular correlates of forms of homeostatic plasticity and potentiation of neurotransmitter release [54–58]. This result matched the possibility that the NMJs evade severe dysfunction through a form of synaptic homeostasis.

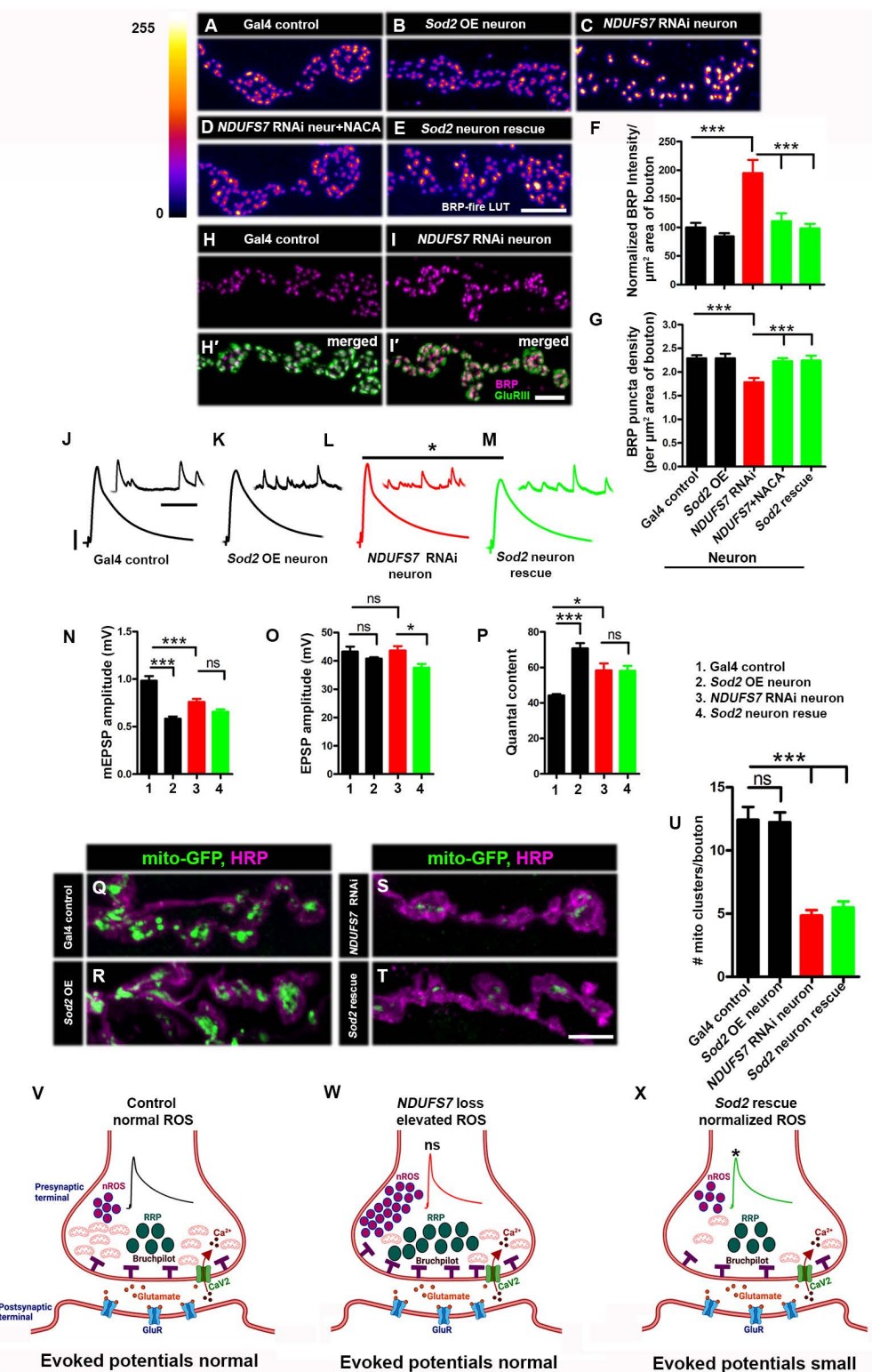

**Fig 4. Neuronal ROS (nROS) controls active zone material levels at NMJs. (A)** Representative images of the A2 hemisegment of muscle 6/7 NMJs in *UAS-mito-GFP, D42-Gal4/+*, **(B)** *UAS-Sod2/+; UAS-mito-GFP, D42-Gal4/+*, **(C)** *NDUFS7[RNAi]/+; UAS-mito-GFP, D42-Gal4/+*, **(D)** *NDUFS7[RNAi]/+; UAS-mito-GFP, D42-Gal4/+* with NACA, and **(E)** *NDUFS7[RNAi]/UAS-Sod2; UAS-mito-GFP, D42-Gal4/+* larvae immunostained with antibodies against

the active zone scaffold Bruchpilot (BRP:fire-LuT) to label the active zones. BRP levels are upregulated at the NMJs in *NDUFS7*-depleted flies, while overexpression of ROS scavenger gene *Sod2* in the neuron or feeding the larvae with N-Acetyl ʟ-cysteine amide (NACA) restores BRP to the control level. (A–E) Scale bar: 2.5 µm. **(F–G)** Histograms showing quantification of BRP intensity (F) and density (G) per µm 2 area of bouton at muscle 6/7 in the genotypes mentioned above. At least 8 NMJs of each genotype were used for quantification. \***p < 0.0001. Error bars denote mean ± s.e.m. Statistical analysis based on one-way ANOVA followed by post-hoc Tukey's multiple-comparison test. (H, H′ and I, I′) Representative confocal images of muscle 6/7 NMJs in the (H, H′) control (*D42-Gal4*/+) and (I, I′) motor neuron Gal4-driven *NDUFS7[RNAi]* (*NDUFS7[RNAi]*/+; *D42-Gal4*/+) immunostained with antibodies against Bruchpilot (BRP: magenta) and GluRIII (green) to label a glutamate receptor subunit. (H, H′ and I, I′) Scale bar: 2.5 µm. There are no significant changes of GluRIII-BRP apposed clusters. At least 8 NMJs of each genotype were used for quantification **(J–P)**. Representative traces, quantification of mEPSPs, EPSPs, and quantal content in the indicated genotypes. Scale bars for EPSPs (mEPSP) are *x* = 50 ms (1,000 ms) and *y* = 10 mV (1 mV). EPSPs amplitudes were maintained in *NDUFS7*-depleted flies due to increased levels of active zone material (e.g., BRP); however, NMJs with *Sod2*-rescued *NDUFS7[RNAi]* in neurons showed diminished evoked release when compared with *NDUFS7[RNAi]*. Minimum 8 NMJs recordings of each genotype were used for quantification. \*p < 0.05, \***p < 0.0001; ns, not significant. Statistical analysis based on one-way ANOVA followed by post-hoc Tukey's multiple-comparison test. Error bars denote the standard error of the mean. **(Q)** Representative images of the A2 hemisegment of muscle 6/7 NMJs in *UAS-mito-GFP, D42-Gal4*/+, **(R)** *UAS-Sod2*/+; *UAS-mito-GFP, D42-Gal4*/+, **(S)** *NDUFS7[RNAi]*/+; *UAS-mito-GFP, D42-Gal4*/+, and **(T)** *NDUFS7[RNAi]*/*UAS-Sod2; UAS-mito-GFP, D42-Gal4*/+ larvae immunostained with antibodies against HRP (magenta) and GFP (mito-GFP:green) to label neurons and mitochondria. *NDUFS7*-depleted and *Sod2*-rescued *NDUFS7*[RNAi] animals harbor fewer mitochondria at the terminals than control animals. (Q–T) Scale bar: 5 µm. **(U)** Histograms showing quantification of mitochondrial clusters in the above-indicated genotypes. **(V–X)** Schematic illustration (drawn on bioRender) showing ROS (magenta) levels, BRP (gray) and mitochondria (red) number in the indicated genotypes. At least 8 NMJs of each genotype were used for quantification. \***p < 0.0001. Error bars represent mean ± s.e.m. Statistical analysis based on one-way ANOVA followed by post-hoc Tukey's multiple-comparison test. Raw data for this figure are available in the S1 Data Excel file, tab Fig 4.

As an independent test, we impaired MCI pharmacologically. To do this, we raised larvae on 50 µM rotenone-spiked food; and we also incubated wild-type fillet preparations with 500 µM of rotenone for extended time. For both cases, we observed significant increases in BRP protein at the presynaptic active zones (S12A–S12U Fig and Table L in S1 Tables). For the extended incubation, the fillet preparations required sufficient rotenone incubation time (six hours) and an intact motor nerve to show the active zone enhancement (S12U Fig and Table L in S1 Tables). This result suggested that downstream of MCI depletion, a compensatory delivery of active zone material required either substantial trafficking time and/or fully intact neuroanatomy.

Next, we checked if the enhanced active zone signal was triggered by excess mitochondrial ROS in motor neurons. Indeed, we found that the *NDUFS7*-depletion active zone enhancements were fully reversed by ROS scavengers, either by raising larvae in food containing NACA or by neuronally expressing *UAS-Sod2* (Fig 4D–4G and Table K in S1 Tables). However, they were not reversed by overexpressing *UAS-Sod1* or *UAS-Catalase* (S13A–S13G Fig and Table M in S1 Tables).

Finally, we assessed synapse function. As with our prior recordings, evoked postsynaptic potentials at the NMJ were not significantly changed by *NDUFS7* depletion in motor neurons. But interestingly, scavenging mitochondrial ROS in the *NDUFS7[RNAi]* neuronal depletion background with *UAS-Sod2* unmasked a small deficit in NMJ excitation, compared to controls (Fig 4J–4P and Table K in S1 Tables). As before, neither *UAS-Sod1* overexpression nor *UAS-Catalase* overexpression unmasked this deficit in NMJ excitation (S13N–S13P Fig and Table M in S1 Tables).

These data could mean that mitochondrial ROS is helping to maintain synaptic activity. Neuronal expression of *UAS-Sod2* did not restore mitochondrial clusters to the NMJ after *NDUFS7* gene function depletion (Fig 4Q–4U and Table K in S1 Tables; like Fig 2H), meaning that the synaptic sites were still deficient in mitochondria. And as expected, neuronal expression of *UAS-Sod1* and *UAS-Catalase* also failed to mitochondrial clusters to the NMJ (S13Q–S13W Fig and Table M in S1 Tables). Collectively, our data support a model in which neuronal ROS (nROS) triggers active zone enhancement and functional compensation when MCI is limiting (Fig 4V–4X and Table K in S1 Tables).

## Neuronal MCI subunits stabilize synaptic strength in conjunction with intracellular calcium signaling proteins

Recent work described a mechanism for local calcium uptake into mitochondria that drives ATP production to maintain synaptic function [59]. The process is governed by the mitochondrial calcium uniporter (MCU) and its accessory EF-hand

MICU proteins [59]. Beyond this role for mitochondrial calcium, there are also known roles for core synaptic functions like vesicle cycling [60,61].

To test if mitochondrial or neuronal calcium could be involved in maintaining synapse function at the NMJ, we acquired genetic reagents to examine depressed MCU function in conjunction with depressed MCI. We also used pharmacological reagents to inhibit release of intracellular sources of calcium, like those from the Ryanodine Receptor (RyR) and the IP$_3$ receptor (IP$_3$R) of the endoplasmic reticulum (ER) [62], as well as a genetic reagent that we previously used at the NMJ to deplete IP$_3$ signaling (*UAS-IP$_3$-sponge*) (Fig 5A–5D). For this set of experiments, we used the *NDUFS7* neuronal knockdown condition as a sensitized genetic background (Fig 5E–5T and Table N in S1 Tables). We also used a pan-neuronal driver (*elaV-Gal4*) to facilitate *Drosophila* genetic stock construction instead of a motor neuronal driver. We previously observed similar NMJ effects either for motor neuronal or pan-neuronal depletion of MCI [13].

We observed no significant differences in EPSP amplitudes when we impaired *mcu* function neuronally (Table N in S1 Tables). Similarly, we did not observe deficits in baseline synaptic activity by blocking RyR and IP$_3$R, alone or in conjunction with *NDUFS7[RNAi]* (Fig 5E–5H and 5Q–5T and Table N in S1 Tables). However, when we concurrently impaired a combination of *NDUFS7*, *mcu*, and those ER calcium store channels, we observed marked decreases in evoked amplitude (Fig 5K, 5L, and 5S and Table N in S1 Tables). These results are consistent with a model in which mitochondrial calcium uptake, MCU activation, and ER (store) calcium efflux combine to stabilize synaptic strength.

If this idea were correct, then it should also be possible to chelate cytoplasmic calcium in a neuronal *NDUFS7[RNAi]* background and reveal neurotransmission defects. Direct application of the membrane-permeable chelator BAPTA-AM, followed by a wash to remove chelator residing in the saline, had no significant effect on neurotransmission parameters versus the mock-treated baseline control (DMSO carrier + wash). But in the *NDUFS7[RNAi]* background, BAPTA-AM + wash significantly diminished evoked potentials, compared to mock-treated (DMSO + wash) NMJs. (Fig 5M–5T and Table N in S1 Tables).

To check if these effects on neurotransmission correlated with effects on active zone protein accumulation, we conducted anti-BRP immunostaining experiments (S14 Fig). As before, neuronal knockdown of *NDUFS7* gene function triggered a marked, compensatory increase in presynaptic active zone material that was readily apparent by confocal microscopy (S14A, S14C, and S14I Fig and Table O in S1 Tables). But this increase was reversed when combined with *mcu* gene function knockdown and pharmacological blockade of store calcium release channels (S14G–S14I Fig and Table O in S1 Tables). Together, our data indicate that loss of MCI subunits in neurons sensitizes synapses to decreases in intracellular calcium.

## A combination of mitochondria, glycolysis, and the TCA Cycle stabilizes NMJ function

Neuronal calcium handling and MCU play roles in NMJ stability that are uncovered by loss of MCI at the NMJ. Downstream of calcium handling, a logical hypothesis is that synaptic energy would play a role [59]. If this were the case, would mitochondria be the sole energy source? Alternatively, in the absence of full mitochondrial function, could glycolysis theoretically substitute, as a homeostatic (or redundant) means for staving off synapse dysfunction?

We tested these ideas by limiting glycolysis as an energy source in two ways: (1) swapping out sucrose and trehalose in our recording saline in favor of 2-deoxy-D-glucose (a nonglycolytic sugar; (Fig 6A); (2) addition of lonidamine (LDA) to the saline to acutely inhibit hexokinase (Fig 6A). Control recordings with these conditions showed little effect on baseline physiology (Fig 6A, 6D, 6F, and 6H–6J and Table P in S1 Tables). However, when Mitochondrial Complex I was impaired neuronally through *NDUFS7[RNAi]* in combination with inhibition of glycolysis, there was a drop in evoked neurotransmission (Fig 6C, 6E, 6G, and 6H–6J and Table P in S1 Tables). This correlated with a failure to increase active zone material after *NDUFS7* gene knockdown (Fig 6K–6Q and Table P in S1 Tables). These results match the idea that a combination of mitochondrial function or glycolysis can work to maintain normal levels of NMJ output.

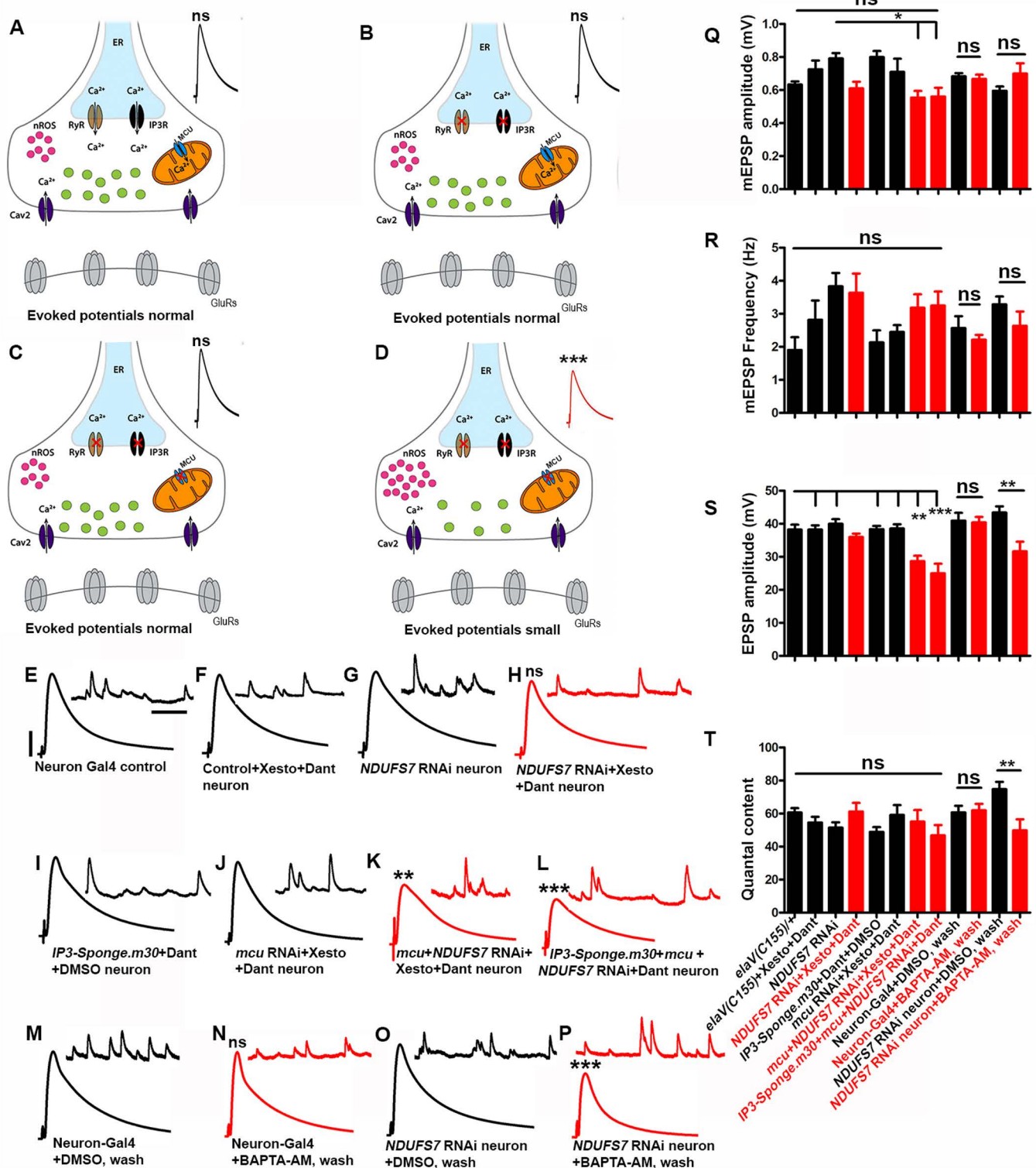

**Fig 5. Loss of an MCI subunit necessitates ER-mediated calcium release to maintain evoked neurotransmission at the NMJs. (A–D)** Schematics (drawn on Adobe Illustrator) illustrating the role of IP$_3$ receptor (IP$_3$R), Ryanodine receptor (RyR) in the endoplasmic reticulum, mitochondrial calcium uniporter Complex (MCU), Ca$_v$2 calcium channel, and synaptic vesicles at the presynaptic nerve terminal. The IP$_3$ and Ryanodine receptors were blocked pharmacologically (red X symbols) by using the IP$_3$R antagonist Xestospongin C or neuronally expressing *UAS-IP$_3$-Sponge* and RyR antagonist

Dantrolene, while *mcu[RNAi]* was used to block the Mitochondrial calcium uniporter complex (red X). **(E–P)** Representative traces of mEPSPs and EPSPs in (E) pan-neuronal Gal4 control (*elaV(C155)-Gal4/+*), (F) pan-neuronal Gal4 control (*elaV(C155)-Gal4/+*) with an acute application (10 min) of 20 µM Xestospongin C and 10 µM Dantrolene, (G) pan-neuronal Gal4-driven *NDUFS7[RNAi]* (*elaV(C155)-Gal4)/+; NDUFS7[RNAi]/+*), (H) pan-neuronal Gal4-driven *NDUFS7[RNAi]* (*elaV(C155)-Gal4/+; NDUFS7[RNAi]/+*) with 20 µM Xestospongin C and 10 µM Dantrolene, (I) pan-neuronal Gal4-driven *UAS-IP3-Sponge.m30* with an acute application of 10 µM Dantrolene (*elaV(C155)-Gal4)/+; NDUFS7[RNAi]/+; UAS-IP3-Sponge.m30/+*), (J) pan-neuronal Gal4-driven *mcu[RNAi]* with 20 µM Xestospongin C and 10 µM Dantrolene (*elaV(C155)-Gal4)/+; mcu[RNAi]/+*), (K) pan-neuronal *mcu[RNAi]* + *NDUFS7*[RNAi] with 20 µM Xestospongin C and 10 µM Dantrolene and(*elaV(C155)-Gal4)/+; mcu[RNAi]/NDUFS7[RNAi]*) and (L) pan-neuronal *UAS-IP3-Sponge.m30* + *mcu[RNAi]* + *NDUFS7*[RNAi] with 10 µM Dantrolene (*elaV(C155)-Gal4/+;NDUFS7[RNAi]/mcu[RNAi];UAS-IP$_3$-Sponge.m30/+*), **(M)** Pan-neuronal Gal4 (*elaV(C155)-Gal4/+*) with DMSO, **(N)** pan-neuronal Gal4 control (*elaV(C155)-Gal4/+*) with 20 µM BAPTA-AM, **(O)** pan-neuronal Gal4-driven *NDUFS7[RNAi]* (*elaV(C155)-Gal4)/+; NDUFS7[RNAi]/+*) with DMSO and **(P)** pan-neuronal Gal4-driven *NDUFS7[RNAi]* (*elaV(C155)-Gal4)/+; NDUFS7[RNAi]/+*) with 20 µM BAPTA-AM. Scale bars for EPSPs (mEPSP) are $x = 50$ ms (1,000 ms) and $y = 10$ mV (1 mV). Note that EPSP amplitudes were reduced in pan-neuronal Gal4-driven *mcu[RNAi]* + *NDUFS7[RNAi]* with an acute exposure of 20 µM Xestospongin C, 10 µM Dantrolene or *UAS-IP$_3$-sponge.m30*+ *mcu[RNAi]* + *NDUFS7*[RNAi] with 10 µM Dantrolene and pan-neuronal Gal4-driven *NDUFS7[RNAi]* with 20 µM BAPTA-AM. **(Q–T)** Histograms showing average mEPSPs, EPSPs amplitude, and quantal content in the indicated genotypes. A minimum of 8 NMJs recordings of each genotype were used for quantification. **$p < 0.05$ (EPSP and QC: *NDUFS7*[RNAi] neuron + DMSO, wash vs. *NDUFS7*[RNAi] neuron + BAPTA-AM, wash), *$p < 0.05$, **$p = 0.001$, ***$p < 0.0001$; ns, not significant. Statistical analysis based on one-way ANOVA followed by post-hoc Tukey's multiple-comparison test. Error bars represent mean ± s.e.m. Raw data for this figure are available in the S1 Data Excel file, tab Fig 5.

We continued this line of investigation genetically. We acquired RNA interference-based transgenes to target five genes involved in *Drosophila* glycolysis or subsequent ATP generation in the Citric Acid (TCA) Cycle: *hexokinase A (hex-A), hexokinase C (hex-C), Citrate (Si) Synthase I, Isocitrate dehydrogenase* (*Idh*), and *Succinyl-coenzyme A synthetase α subunit 1* (*Scsα1*). We knocked down these genes neuronally, either alone or in combination with *NDUFS7[RNAi]* (S15 and S16 Figs). Neuronal impairment of *hex-C* had no effect on baseline neurotransmission, but *hex-A* impairment reduced it (S15A–S15E Fig and Table Q in S1 Tables). Impairment of the TCA Cycle enzymes on their own had little-to-no effect on baseline neurotransmission (S15A and S15F–S15H and Table Q in S1 Tables). However, concurrent impairment of *NDUFS7* and most of these genes significantly blunted neurotransmission, with the exception being *hex-C* (S15I–S15N Fig and Table Q in S1 Tables). Collectively, the data suggest that MCI works in conjunction with—or redundantly to—alternative energy-generating pathways to support normal levels of neurotransmission (S15O–S15Q Fig). In the case of the hexokinases, *Drosophila hex-A* seems more important for this process than *hex-C*.

We also quantified active zone material accumulation. The results mirrored the neurotransmission tests: as before, neuronal *NDUFS7[RNAi]* impairment elicited enhanced active zone material (S16A, S16B, and S16M and Table R in S1 Tables). On their own, impairment of glycolysis or TCA Cycle genes had variable effects active zone material (S16C–S16G and S16M Fig and Table R in S1 Tables). But concurrent neuronal impairment of *NDUFS7* and any of the glycolysis or TCA Cycle genes reversed active zone enhancement (S16H–S16M Fig and Table R in S1 Tables).

## MCI subunits in muscle are required for proper synapse development

We previously reported that impairments of MCI diminish *Drosophila* NMJ growth [13]. The roles of MCI in specific tissues for this developmental process were unclear. For the present study, we tested for tissue-specific roles of MCI in NMJ development. To visualize NMJ boutons, we co-stained larval fillets with anti-Horseradish Peroxidase (HRP) a presynaptic membrane marker, and anti-Discs Large (Dlg), a postsynaptic density marker [63,64].

On a coarse level, MCI loss in muscle (*BG57-Gal4 > UAS-NDUFS7[RNAi]*) caused a severe reduction in average bouton size, a decrease in bouton number, a notable decrease in Dlg expression, and a bouton "clustering" phenotype (Fig 7A and 7B), reminiscent of what we previously reported [13]. To quantify these observations, we measured bouton number, muscle area, and branch number per muscle in the third-instar larval NMJ synapses (Table S in S1 Tables). We found that *NDUFS7* muscle knockdown resulted in a significant reduction in all these parameters compared to controls (Fig 7J–JL, 7U, and 7V and Table S in S1 Tables).

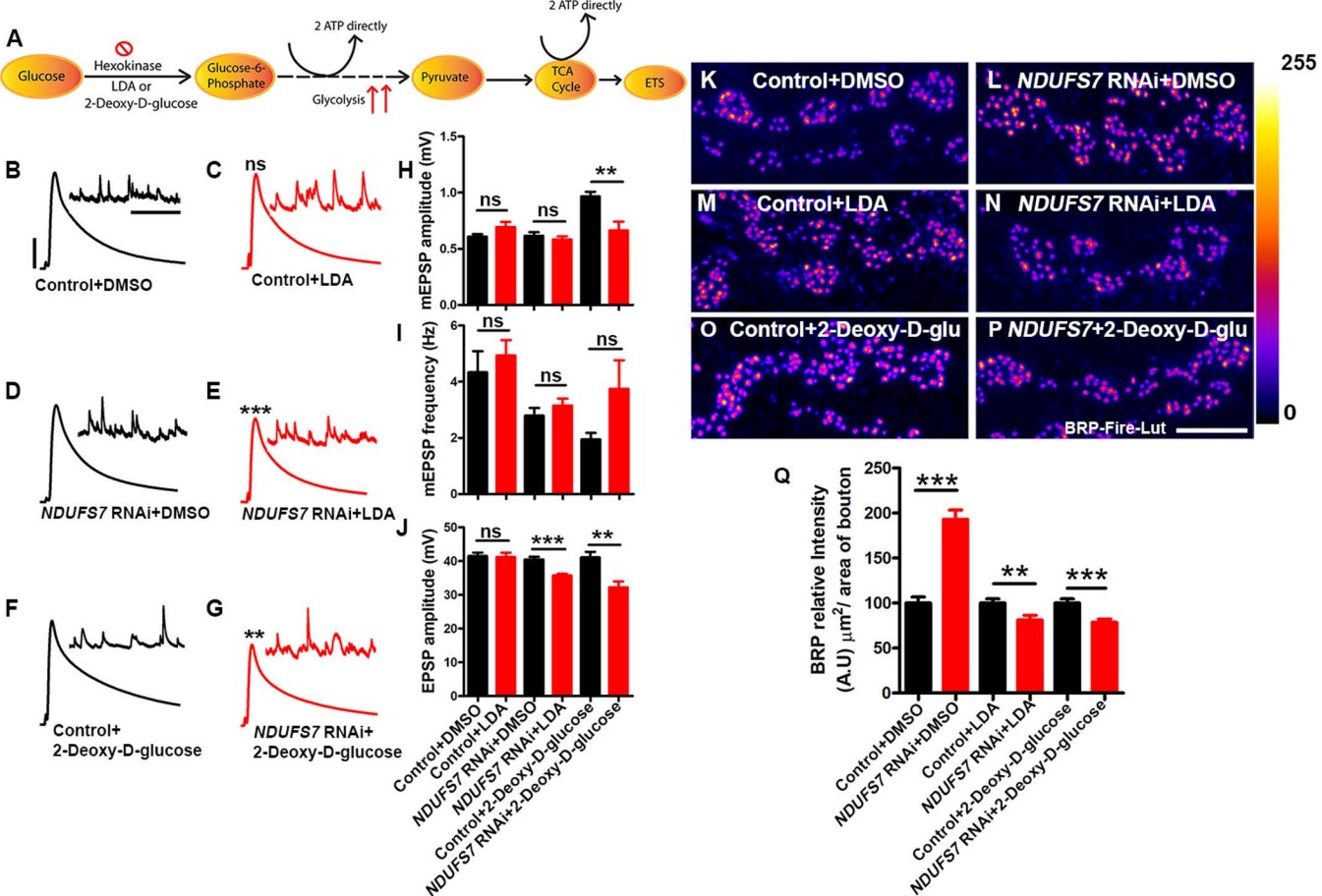

**Fig 6. Loss of an MCI subunit necessitates glycolysis to regulate levels of active zone materials and stabilize synaptic strength. (A)** Schematic illustrations showing steps of glucose metabolism and ATP production during glycolysis and TCA cycle in the cell. **(B–G)** Representative traces of mEPSPs and EPSPs in (B) pan-neuronal Gal4 control (*elaV(C155)-Gal4*/+ with DMSO), (C) pan-neuronal Gal4 control (*elaV(C155)-Gal4*/+) with an acute application (30 min) of 150 μM Lonidamine (LDA) (D) pan-neuronal Gal4-driven *NDUFS7[RNAi]* (*elaV(C155)-Gal4)/+; NDUFS7[RNAi]*/+ with DMSO, (E) pan-neuronal Gal4-driven *NDUFS7[RNAi]* (*elaV(C155)-Gal4)/+; NDUFS7[RNAi]*/+) with 150 μM Lonidamine, (F) pan-neuronal Gal4 control (*elaV(C155)-Gal4*/+) and HL3 containing 2-Deoxy-ᴅ-glucose and (G) pan-neuronal Gal4-driven *NDUFS7[RNAi]* (*elaV(C155)-Gal4)/+; NDUFS7[RNAi]*/+) with recording saline containing 2-Deoxy-ᴅ-glucose as the sugar source. Scale bars for EPSPs (mEPSP) are $x = 50$ ms (1,000 ms) and $y = 10$ mV (1 mV). The EPSPs amplitudes were reduced in pan-neuronal Gal4-driven *NDUFS7[RNAi]* with an acute exposure of 150 μM Lonidamine (LDA) or pan-neuronal Gal4-driven *NDUFS7[RNAi]* in 2-Deoxy-ᴅ-glucose for 30 min. There was also a reduction in mEPSP amplitudes in saline containing 2-Deoxy-ᴅ-glucose in pan-neuronal Gal4-driven *NDUFS7-depleted* larvae compared to the control animals. **(H–J)** Histograms showing average mEPSPs, EPSPs amplitude, and frequencies in the indicated genotypes. A minimum of 8 NMJ recordings of each genotype were used for quantification. **p = 0.003 (mEPSP amplitude: *elaV(C155)-Gal4*/+2-Deoxy-ᴅ-glucose vs *elaV(C155)-Gal4)/+; NDUFS7[RNAi]*/+ with 2-Deoxy-ᴅ-glucose), **p=0.002 (EPSP amplitude: *elaV(C155)-Gal4*/+ with 2-Deoxy-ᴅ-glucose vs. *elaV(C155)-Gal4)/+; NDUFS7[RNAi]*/+ with 2-Deoxy-ᴅ-glucose),***p=0.0001 (EPSP: *elaV(C155)-Gal4*/+ with LDA vs *elaV(C155)-Gal4)/+; NDUFS7[RNAi]*/+ with LDA); ns, not significant. Statistical analysis is based on the Student t test for pairwise sample comparison. Error bars represent mean±s.e.m. **(K–P)** Representative images of the A2 hemisegment of muscle 6/7 NMJs in the above-indicated genotypes immunostained with antibodies against the active zone scaffold Bruchpilot (BRP:fire-LuT) to label the active zones. The BRP levels downregulated at the NMJs in pan-neuronal Gal4-driven *NDUFS7*[RNAi] either incubated with 150 μM LDA or in HL3 containing 2-Deoxy-ᴅ-glucose for 30 min. (K–P) Scale bar: 5 μm. **(Q)** Histograms showing quantification of BRP intensity in μm 2 area of bouton at muscle 6/7 in the genotypes mentioned above. At least 8 NMJs of each genotype were used for quantification.***p < 0.0001 (BRP levels: *elaV(C155)-Gal4*/+ with DMSO vs *elaV(C155)-Gal4)/+; NDUFS7[RNAi]*/+ with DMSO),**p=0.008 (BRP levels: *elaV(C155)-Gal4*/+ with LDA vs. *elaV(C155)-Gal4)/+; NDUFS7[RNAi]*/+ with LDA), ***p=0.0005 (BRP levels: *elaV(C155)-Gal4*/+ with 2-Deoxy-ᴅ-glucose vs *elaV(C155)-Gal4)/+; NDUFS7[RNAi]*/+ with 2-Deoxy-ᴅ-glucose). Error bars denote mean±s.e.m. Statistical analysis based on one-way ANOVA followed by post-hoc Tukey's multiple-comparison test. Raw data for this figure are available in the S1 Data Excel file, tab Fig 6.

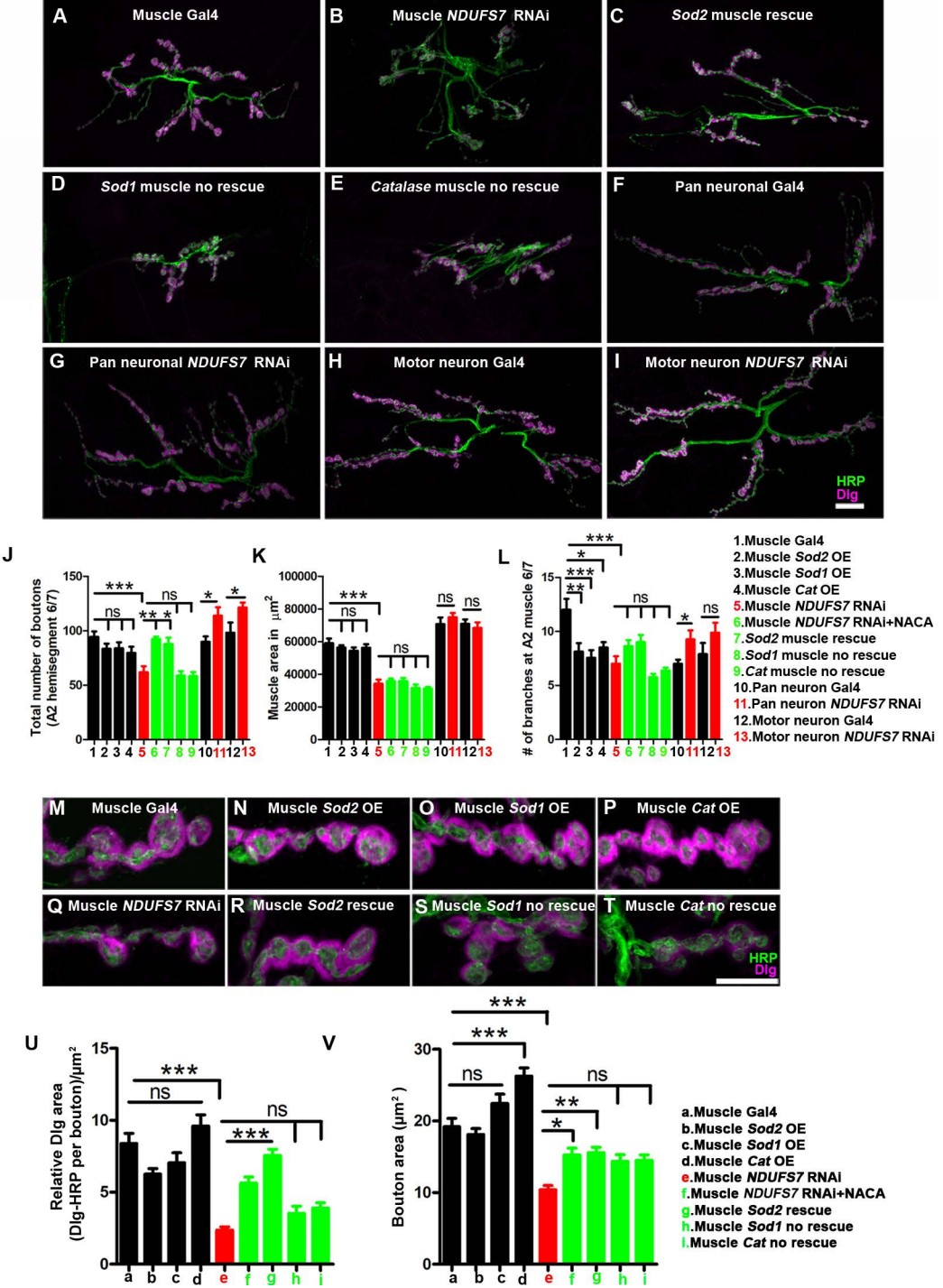

**Fig 7. NDUFS7 in muscles is required to promote normal synapse growth.** (A–I) Representative confocal images of NMJ synapses at muscle 6/7 of **(A)** Muscle-Gal4 control (*BG57-Gal4/+*), **(B)** Muscle Gal4-driven *NDUFS7[RNAi]* (*NDUFS7[RNAi]/+; BG57-Gal4/+*), **(C)** *Sod2* muscle rescue (*UAS-Sod2/NDUFS7[RNAi]; BG57-Gal4/BG57-Gal4*), **(D)** *Sod1* muscle nonrescue (*UAS-Sod1/NDUFS7[RNAi]; BG57-Gal4/BG57-Gal4*), **(E)** Catalase muscle nonrescue (*UAS-Cat/NDUFS7[RNAi]; BG57-Gal4/BG57-Gal4*), **(F)** Pan-neuronal Gal4 control (*elaV(C155)-Gal4)/+*), (G) *elaV(C155)-Gal4*-driven *NDUFS7[RNAi]* (*elaV(C155)-Gal4/+; NDUFS7[RNAi]/+*), **(H)** Motor neuron Gal4 control (*D42-Gal4/+*), and **(I)** *D42*-Gal4-driven *NDUFS7[RNAi]* (*NDUFS7[RNAi]/+; D42-Gal4/+*) double immunolabeled with Dlg (magenta) and HRP (green) antibodies. (A–I) Scale bar: 10 μm. The NMJ morphological defects in *NDUFS7[RNAi]* were restored upon co-expression of *UAS-Sod2 in* muscle; however, it did not rescue with *UAS-Sod1* or *UAS-Catalase*

transgene. **(J–L)** Histograms show the number of boutons, muscle area, and average NMJ length at muscle 6/7 of A2 hemisegment in the indicated genotypes. *$p < 0.05$, *$p = 0.006$ (# of boutons: Muscle *NDUFS7[RNAi]* vs. *Sod2* muscle rescue), **$p = 0.0002$ (# of boutons: Muscle *NDUFS7*[RNAi] vs. Muscle *NDUFS7[RNAi]* +NACA), *$p = 0.009$ (# of branches: Muscle Gal4 vs. Muscle *Sod2* OE), ***$p = 0.002$ (# of branches: Muscle Gal4 vs. Muscle *Sod1* OE), *$p = 0.008$ (# of branches: Muscle Gal4 vs. Muscle *cat* OE), ***$p = 0.001$ (Muscle Gal4 vs. Muscle *NDUFS7*[RNAi]), ***$p < 0.0001$; ns, not significant. Statistical analysis based on one-way ANOVA with post-hoc Tukey's test for multiple and Student $t$ tests for pairwise comparison. Error bars represent mean ± s.e.m. **(M–T)** Representative confocal images of boutons at the third instar larval NMJ synapse in **(M)** Muscle-Gal4 control (*BG57-Gal4/+*), **(N)** Muscle Gal4-driven *UAS-Sod2* (*UAS-Sod2/+; BG57-Gal4/+*), **(O)** *UAS-Sod1* (*UAS-Sod1/+; BG57-Gal4/+*), **(P)** *UAS-Catalase* (*UAS-Catalase/+; BG57-Gal4/+*) **(Q)** Muscle *NDUFS7[RNAi]* (*NDUFS7[RNAi]/+; BG57-Gal4* **(R)** *Sod2* muscle rescue (*UAS-Sod2/NDUFS7[RNAi]; BG57-Gal4/BG57-Gal4*), **(S)** *Sod1* muscle rescue (*UAS-Sod1/NDUFS7[RNAi]; BG57-Gal4/BG57-Gal4*), and **(T)** Catalase muscle rescue (*UAS-Cat/NDUFS7[RNAi]; BG57-Gal4/BG57-Gal4*) animals double immunolabeled with anti-HRP (green) and anti-Dlg (magenta) antibodies. (M–T) Scale bar: 5 µm. Note that the gross morphology of SSRs and immunoreactivity of Dlg were reduced in *NDUFS7[RNAi]* animals. As mentioned, phenotypes were restored to wild-type level when *NDUFS7*[RNAi]-depleted flies were reared in NACA or by genetically expressing *Sod2 transgene* in muscle. **(U, V)** Histograms showing normalized synaptic fluorescence of Dlg and bouton area in the indicated genotypes. *$p < 0.01$, **$p < 0.001$, ***$p < 0.0001$; ns, not significant. Error bars represent mean ± s.e.m. Statistical analysis based on one-way ANOVA with post-hoc Tukey's test for multiple and Student $t$ tests for pairwise comparison. Raw data for this figure are available in the S1 Data Excel file, tab Fig 7.

In contrast to muscle knockdown, pan-neuronal or motor neuron-specific knockdown of *NDUFS7* showed slight NMJ overgrowth phenotypes (Fig 7F–7L and Table S in S1 Tables). As was the case with the active zone enhancement associated with neuronal *NDUFS7* knockdown (Fig 4), this NMJ overgrowth could represent a developmental mechanism to stave off dysfunction caused by missing neuronal mitochondria. Taken together, our NMJ immunostaining results indicated to us that blunted NMJ growth due to MCI loss was likely due to muscle MCI dysfunction.

We tested if the NMJ undergrowth phenotypes could be due to the increased levels of mitochondrial ROS we had observed in muscle (mROS) (S2F Fig). If that idea were correct, then the undergrowth phenotypes should be reversed if mitochondrial mROS were scavenged. Consistent with this idea, bouton number and synaptic undergrowth phenotypes were fully restored to wild-type levels when the *NDUFS7[RNAi]* muscle knockdown animals also had *UAS-Sod2* transgenically expressed in the muscles (Fig 7C, 7J–7L, and 7R and Table S in S1 Tables). They were also restored to wild-type levels when muscle *NDUFS7* knockdown animals were raised on food containing the antioxidant N-acetyl cysteine amide (NACA) (Fig 7J–7L and Table S in S1 Tables). By contrast, none of these NMJ growth parameters were restored to wild-type levels if scavengers *UAS-Sod1* or *UAS-Catalase* were misexpressed in the muscle (Fig 7D–7E and 7J–7L and Table S in S1 Tables).

## Loss of MCI and mROS in muscle disorganize NMJ postsynaptic densities

The muscle Discs Large (Dlg)-Spectrin network functions as an organizing scaffold for synaptic assembly [64,65]. Dysregulation of this network can lead to an aberrant muscle subsynaptic reticulum (SSR) [64]. Therefore, one possible target of excess mROS in the absence of MCI function is Dlg. Dlg is the fly homolog of PSD-95/SAP97/PSD-93, and it is a member of the membrane-associated guanylate kinase (MAGUK) family of NMJ scaffolding proteins [64]. It is present both within presynaptic boutons and in the portion of the SSR closest to the bouton.

Using the same antibodies detailed above (anti-Dlg and anti-HRP), we used a high magnification to examine at the postsynaptic densities closely. By confocal immunofluorescence, Dlg area was significantly reduced when *NDUFS7* was depleted postsynaptically by[RNAi] (Fig 7M–7U and Table S in S1 Tables). To quantify the relative Dlg area, we measured Dlg area with respect to HRP (Relative Dlg area = Dlg area minus HRP area) in type 1b boutons at muscle 6/7 of the A2 hemisegment (Fig 7U). Compared with the control synapses, *NDUFS7* knockdown resulted in a significant reduction in the relative Dlg area (Fig 7M–7U and Table S in S1 Tables). Consistently, the relative α-Spectrin area was also reduced when *NDUFS7* was depleted in muscle (S17A–S17F Fig and Table G in S1 Tables).

Next, we scavenged mROS to check how that affected the postsynaptic densities. The relative Dlg and α-Spectrin areas were restored to wild-type levels when animals were grown in a media containing NACA (*NDUFS7[RNAi]/+; BG57-Gal4/+* with NACA) or genetically expressing *UAS-Sod2* in the muscle (*UAS-Sod2/NDUFS7[RNAi]; BG57-Gal4/*

*BG57-Gal4*) (Dlg data: Fig 7R and 7U and Table S in S1 Tables; α-Spectrin data: S17D–S17F Fig and Table G in S1 Tables). By contrast, Dlg levels were not restored while expressing other scavengers in the muscle, encoded by *UAS-Sod1* and *UAS-Catalase* transgenes (Fig 7S–7U and Table S in S1 Tables). We conclude that depletion of MCI subunits in the muscle disables postsynaptic density formation via the formation of mitochondrial ROS intermediates.

The postsynaptic density (PSD-95/Dlg) has been shown to cluster glutamate receptors at the SSR [66]. Our results raised the possibility that glutamate receptor clusters could be disrupted when *NDUFS7* was depleted in the muscle (Fig 8). To test this idea, we simultaneously immunostained NMJs with antibodies against BRP (neuron, presynaptic active zone) and glutamate receptor clusters (muscle). In controls, these pre and postsynaptic structures were directly apposed to one another (Fig 8A and 8G). But when *NDUFS7* gene function was depleted, we observed "missing" GluRIIA and GluRIII receptor clusters (Fig 8B, 8F, 8H, and 8K and Table G in S1 Tables), i.e., BRP puncta without apposed

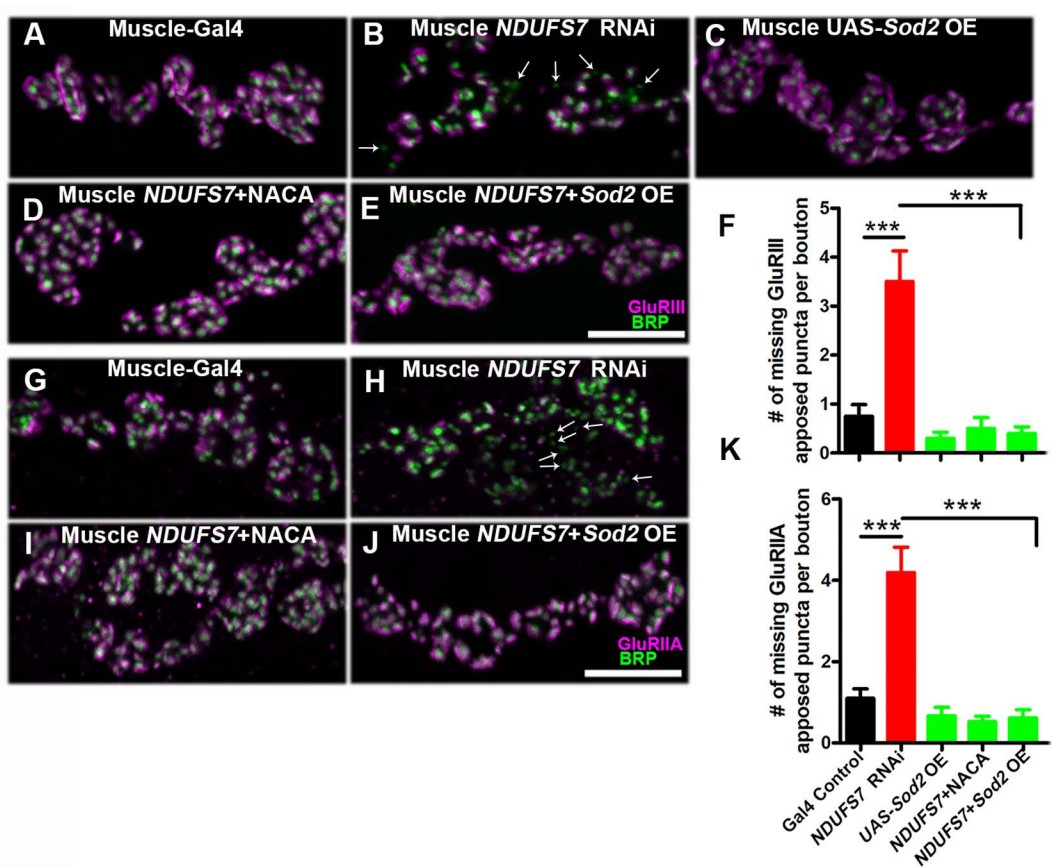

**Fig 8. *NDUFS7* subunit in muscle affects the organization of GluRs cluster in *Drosophila*.** Representative confocal images of boutons at the third instar larval NMJ synapse for **(A)** Muscle-Gal4 control (*BG57-Gal4*/+), **(B)** Muscle *NDUFS7[RNAi]* (*NDUFS7[RNAi]*/+; *BG57-Gal4*/+), **(C)** Muscle Gal4-driven *UAS-Sod2* (*UAS-Sod2*/+; *BG57-Gal4*/+), **(D)** *NDUFS7[RNAi]*/+; *BG57-Gal4*/+NACA), and **(E)** *Sod2* muscle rescue (*UAS-Sod2*/*NDUFS7[RNAi]*; *BG57-Gal4*/*BG57-Gal4*) animals immunolabeled with active zone marker BRP (green) and anti-GluRIII (magenta) antibodies. Scale bar: 5 μm. Note that GluRIII apposed clusters with BRP are missing in the *NDUFS7[RNAi]* knockdown animals (marked in arrow) compared to control. These phenotypes were restored to normal when *NDUFS7[RNAi]* knockdown flies were reared in media containing NACA or genetically expressing *Sod2 transgene* in muscle. **(F)** Histograms showing quantification of the number of missing BRP-GluRIII apposed puncta per bouton in the indicated genotypes. **(G–J)** Similar phenotypes were observed when analyzed for BRP-GluRIIA apposed clusters in boutons. **(K)** Histograms showing quantification of the number of missing BRP-GluRIIA apposed puncta per bouton in the indicated genotypes. ***$p < 0.0001$. Error bars represent mean ± s.e.m. Statistical analysis based on one-way ANOVA with post-hoc Tukey's test for multiple comparisons. Raw data for this figure are available in the S1 Data Excel file, tab Fig 8.

glutamate receptors. These lack of apposition phenotypes were fully reversed by raising the larvae with NACA or genetically expressing *UAS-Sod2* in the muscle (Fig 8D–8F and 8I–8K and Table G in S1 Tables). However, as with prior tests, these phenotypes were not reversed by *UAS-Sod1* or *UAS-Catalase* overexpression (S18 Fig and Table H in S1 Tables). Together, our data indicate that loss of MCI subunit in the muscle (*NDUFS7*) disrupts several aspects of the postsynaptic density organization, and these disruptions are likely due to the accumulation of mitochondrial ROS in the muscle.

**Loss of MCI subunits in muscle diminishes evoked NMJ neurotransmission through postsynaptic mROS**

Mitochondria and ROS influence presynaptic vesicle release and plasticity at synapses. This has been shown in diverse model systems like flies, mice, and worms [52,53,67–69]. In our prior work, we identified an NMJ neurotransmission defect when MCI function is impaired [13], but based on our current study, that defect does not seem like it is dependent upon MCI's neuronal functions (Figs 1 and 4–6). Therefore, we turned to analyzing postsynaptic muscle MCI and mROS to check if these parameters influenced neurotransmission.

We performed sharp electrode electrophysiological recordings of miniature and excitatory postsynaptic potentials (mEPSP and EPSP) at NMJ muscle 6, hemisegment A2. We also used average mEPSP and EPSP values to estimate QC for each NMJ. For the most part, mEPSP values remained steady (Fig 9A–9E and Table T in S1 Tables), with some exceptions. The starkest phenotypes came in terms of evoked amplitudes (Fig 9A–9D and 9F and Table T in S1 Tables), sometimes due to changes in QC (Fig 9G) or combinatorial changes in both mEPSP (Fig 9E) and QC (Fig 9G).

Specifically, evoked synaptic vesicle release (EPSP) was significantly reduced when *NDUFS7* was depleted in muscles by RNAi (Fig 9A and 9F and Table T in S1 Tables). That reduction was reversed after scavenging mitochondrial ROS in the muscle. Indeed, transgenic muscle-driven *Sod2* expression suppressed the *NDUFS7[RNAi]* phenotype, but expression of the *Catalase* and *Sod1* did not (Fig 9A and 9F and Table T in S1 Tables). We also tested if feeding a ROS scavenger to developing larvae would reverse the same neurotransmission defect. Carrier feeding alone (10% EtOH) did not affect evoked neurotransmission, nor did it influence the neurotransmission loss caused by *NDUFS7[RNAi]* (Fig 9B and 9F and Table T in S1 Tables). Feeding larvae 0.5 mM NACA successfully reversed the phenotype, but the nonspecific additive curcumin had no effect (Fig 9B and 9F and Table T in S1 Tables).

We checked different MCI manipulations. First, we examined hemizygous *ND-30^{EY03664/Df}* genetic mutants. As with muscle-driven *NDUFS7[RNAi]*, the *ND-30* mutant NMJs had blunted evoked neurotransmission, but this defect was successfully reversed by ROS scavengers (Fig 9C and Table T in S1 Tables). We also impaired MCI pharmacologically, by feeding larvae 50 μM rotenone (or 0.5% DMSO carrier control), similar to conditions we previously published [13]. As with the prior study, rotenone blunted neurotransmission (Fig 9D, 9F, and 9G and Table T in S1 Tables), but this effect was ameliorated by a genetic background overexpressing *UAS-Sod2* in the muscle (Fig 9D, 9F, and 9G and Table T in S1 Tables). By contrast, overexpressions of *UAS-Catalase* and *UAS-Sod1* were not effective at reversing the effect of rotenone (Fig 9D, 9F, and 9G and Table T in S1 Tables).

Finally, we performed behavioral experiments on *NDUFS7[RNAi]* and *ND-30[RNAi]* and mutant animals. Consistent with the electrophysiological recordings, MCI muscle-depleted and mutant animals showed severe defects in crawling ability. The crawling behavior was rescued by expressing the *UAS-Sod2 transgene* in the muscle or raising the larvae in a media containing NACA (Fig 9H and Table U in S1 Tables). By contrast, neuronal depletion of *NDUFS7* in larvae did not show any significant crawling defects (Fig 9H and Table U in S1 Tables). Together, our data suggest that excess mitochondrial ROS accumulation in muscle (mROS) diminishes baseline synaptic physiology when MCI activity is lost, and it also triggers aberrant crawling behavior.

## Discussion

We uncovered novel aspects of NMJ synapse biology controlled by Mitochondrial Complex I (MCI). Impairment of MCI causes profound cytological phenotypes in synaptic tissues (Figs 1–3) [13]. By examining mitochondria directly, we

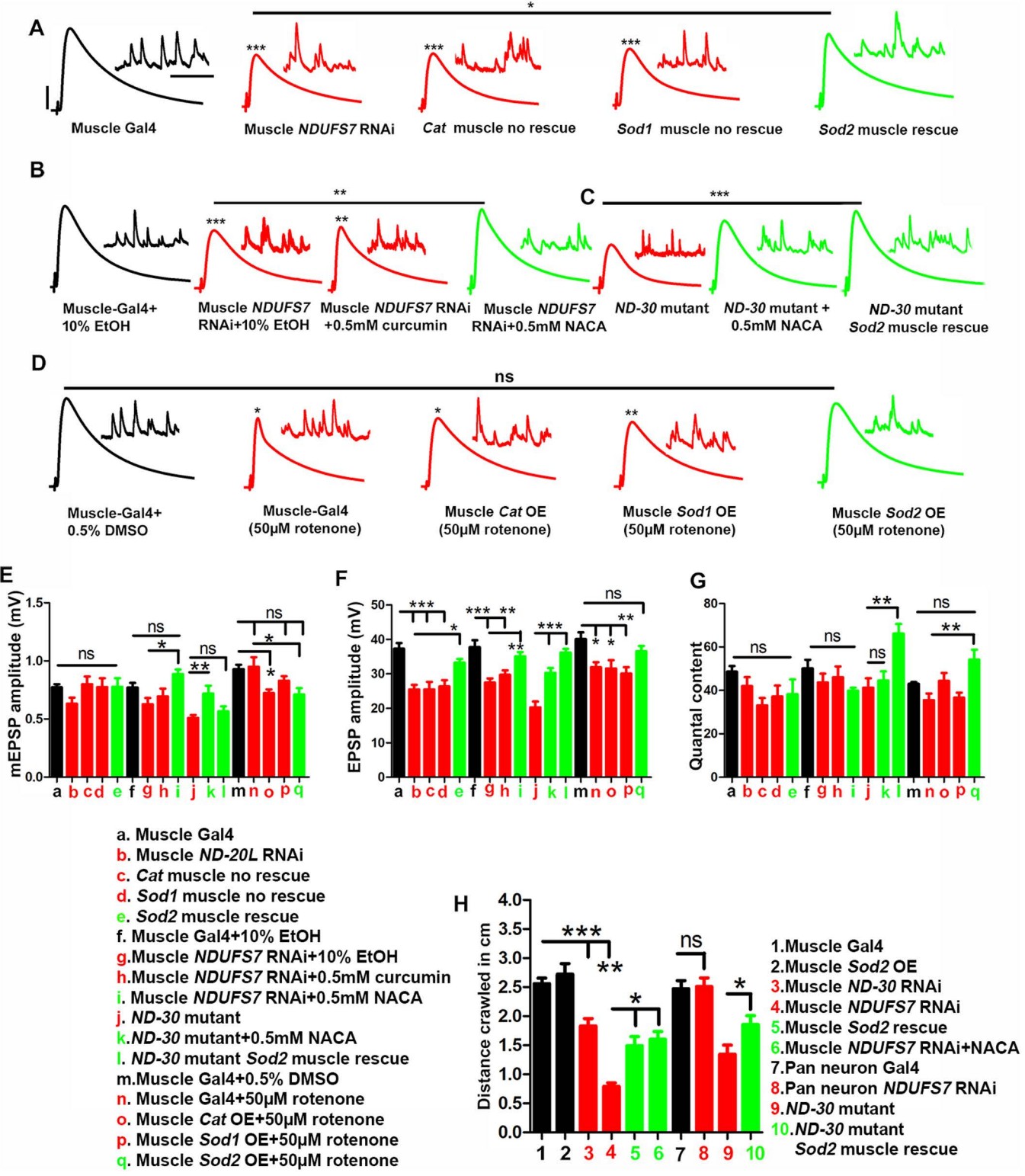

**Fig 9. Loss of MCI subunits affects synaptic transmission via the formation of excess ROS in the muscle. (A)** Representative traces of mEP-SPs and EPSPs in muscle-Gal4 control (*BG57-Gal4*/+), muscle Gal4-driven *NDUFS7[RNAi]* (*NDUFS7[RNAi]*/+; *BG57-Gal4*/+), muscle *Catalase* nonrescue (2× muscle-Gal4>*UAS-Catalase*/*NDUFS7*[RNAi]: *UAS-Catalase*/*NDUFS7[RNAi]; BG57-Gal4/BG57-Gal4*), muscle *Sod1* nonrescue (2×

muscle-Gal4>*UAS-Sod1/NDUFS7*[RNAi]: *UAS-Sod1/NDUFS7[RNAi]; BG57-Gal4/BG57-Gal4*), muscle *Sod2* rescue (2× muscle-Gal4>*UAS-Sod2/NDUFS7*[RNAi]: *UAS-Sod2/NDUFS7[RNAi]; BG57-Gal4/BG57-Gal4*) animals. Note that EPSP amplitudes were reduced in *NDUFS7[RNAi],* and the phenotype was restored to wild-type levels by expressing *Sod2 transgene* in the muscle but not with *Sod1* and *Catalase* transgenes. **(B)** Representative traces of mEPSPs and EPSPs in muscle Gal4 control (*BG57-Gal4*/+) larvae raised on 10% EtOH, *NDUFS7* muscle[RNAi] (*NDUFS7[RNAi]*/+; *BG57-Gal4*/+) raised on 10% EtOH, *NDUFS7* muscle[RNAi] (*NDUFS7[RNAi]*/+; *BG57-Gal4*/+) raised on 0.5mM curcumin, *NDUFS7* muscle[RNAi] (*NDUFS7[RNAi]*/+; *BG57-Gal4*/+) raised on 0.5mM NACA, **(C)** Representative traces of mEPSPs and EPSPs in *ND-30$^{epgy}$/Df* mutants, *ND-30$^{epgy}$/Df* mutants raised on 0.5mM NACA and *UAS-Sod2* muscle rescued *ND-30$^{epgy}$/Df* mutant (*UAS-Sod2/+; ND-30 (Df), BG57-Gal4/ND-30$^{epgy}$*) animals. The EPSPs amplitudes were restored to wild-type when muscle depleted *NDUFS7[RNAi]* larvae were raised in food containing NACA or by *UAS-Sod2* muscle overexpression in *NDUFS7[RNAi]* depleted animals. **(D)** Representative traces of mEPSPs and EPSPs in muscle-Gal4 control (*BG57-Gal4*/+) larvae raised on DMSO, muscle Gal4 (*BG57-Gal4*/+) larvae raised on 50 μM rotenone (complex I inhibitor), *UAS-Catalase* (*UAS-Catalase*/+; *BG57-Gal4*/+), *UAS-Sod1* (*UAS-Sod1*/+; *BG57-Gal4*/+), and *UAS-Sod2* (*UAS-Sod2*/+; *BG57-Gal4*/+) muscle overexpression animals raised on 50 μM rotenone. The EPSP amplitudes were restored in rotenone raised larvae overexpressing *UAS-Sod2* in muscle (*UAS-Sod2*/+; *BG57-Gal4*/+), likely due to its free radical scavenging activity. Scale bars for EPSPs (mEPSP) are $x = 50\,ms$ (1,000 ms) and $y = 10$ mV (1 mV). **(E–G)** Histograms showing average mEPSPs, EPSPs amplitude, and quantal content in the indicated genotypes. Minimum 8 NMJs recordings of each genotype were used for quantification. **(H)** Histogram representing crawling behavior (in cm) of the larvae in the indicated genotypes. Knocking down *NDUFS7[RNAi]* or *ND-30[RNAi]* in muscle and *ND-30$^{epgy}$/Df* mutants showed a severe defect in crawling behavior. The abnormal crawling behavior was rescued by expressing a *Sod2 transgene* in the muscle or rearing the larvae in a media containing NACA. Moreover, neuronally depleting *NDUFS7* did not show any notable change in crawling defects. Minimum 10 animals were analyzed for crawling behavioral analysis. *$p < 0.05$ (mEPSP amplitude: Muscle *NDUFS7[RNAi]* + 10% EtOH vs. Muscle *NDUFS7[RNAi]* + 0.5mM NACA),**$p = 0.006$ (mEPSP amplitude: *ND-30* mutant vs. *N-30* mutant + 0.5mM NACA), *$p = 0.0004$ (mEPSP amplitude: Muscle-Gal4 + 0.5% DMSO vs. Muscle *Cat* OE + 50 μM rotenone),*$p = 0.039$ (mEPSP amplitude: Muscle Gal4 + 50mM rotenone vs. Muscle *Sod2* OE + 50 μM rotenone), *$p = 0.001$ (EPSP amplitude: Muscle *NDUFS7[RNAi]* vs. *Sod2* muscle rescue),**$p = 0.003$ (EPSP amplitude: Muscle Gal4 + 10% EtOH vs. Muscle *NDUFS7[RNAi]* + 0.5mM curcumin), **$p = 0.0004$ (EPSP amplitude: Muscle *NDUFS7[RNAi]* + 10% EtOH vs. Muscle *NDUFS7[RNAi]* + 0.5mM NACA), *$p = 0.004$ (EPSP amplitude: Muscle Gal4 + 0.5% DMSO vs. Muscle Gal4 + 50 μM rotenone), *$p = 0.015$ (EPSP amplitude: Muscle-Gal4 + 0.5% DMSO vs. Muscle *cat* OE + 50 μM rotenone), **$p = 0.015$ (EPSP amplitude: Muscle-Gal4 + 0.5% DMSO vs. Muscle *Sod1* OE + 50 μM rotenone), **$p = 0.001$ (QC: *ND-30* mutant vs. *Sod2* muscle rescue), **$p = 0.003$ (QC: Muscle Gal4 + 50 μM rotenone vs. Muscle *Sod2* OE + 50 μM rotenone), **$p = 0.0002$ (Distance crawled: Muscle Gal4 vs. Muscle *ND-30*[RNAi]), ***$p < 0.0001$; ns, not significant. Statistical analysis based on one-way ANOVA with post-hoc Tukey's test for multiple and Student *t* test for pairwise comparison. Raw data for this figure are available in the S1 Data Excel file, tab Fig 9.

discovered shared phenotypes between MCI loss and loss of the *Drosophila* Mitofusin, Marf (Fig 2). Additionally, with MCI loss, we noted an enhancement of mitochondrial ROS, S2 Fig), consistent with prior work [36–40].

Unexpectedly, these perturbations spur functionally opposite responses in presynaptic neurons versus postsynaptic muscles. In motor neurons, MCI loss and mitochondrial ROS appear to trigger a compensatory response, where the underlying cytological problems are offset by an increase in active zone material, resulting in normal levels of evoked excitation (Fig 4). This process requires known intracellular calcium signaling components (Fig 5). It also appears to require energy stores because loss of glycolysis—which may function as a supplemental energy source to mitochondria— abrogates the presynaptic compensation (Fig 6). By contrast, in the muscle, MCI loss and mitochondrial ROS trigger a destructive response, where there is a disassembly of the postsynaptic density (Fig 7). This disassembly correlates with mis-apposition of pre and postsynaptic structures (Fig 8) and defects in neurotransmission and locomotion (Fig 9).

### Disruption of mitochondrial dynamics, ROS, and physiological responses in *Drosophila*

Energy is needed for normal levels of synaptic transmission [2]. Intuitively, a loss of synaptic mitochondria should blunt transmission because transmission requires energy. Consistently, several labs have previously implicated mitochondrial dynamics in *Drosophila* synapse function, including mitochondrial fission (Dynamin-related protein 1, Drp1 [51]), fusion (Mitofusin/dMarf [31]), trafficking (Miro and Milton [50,70,71]), or quality control (Pink and Parkin [72–75]). Additionally, it has been established from model organisms such as flies, worms, and mice that any misregulation in mitochondrial distribution could affect synaptic activity [51,59,76,77]. Adding to that work, we uncovered synaptic transmission and developmental phenotypes after depletion of MCI at the NMJ [13] potentially related to defective mitochondrial fusion (this study).

Additionally, ROS has previously been studied in the context of mitochondrial dysfunction [78]. Excess ROS can trigger mitochondrial calcium uptake and subsequently trigger apoptosis or degeneration of neurons or neural support cells

[79,80]. MCI deficiency elevates ROS levels, and this process can promote the fragmentation of mitochondria in cells like fibroblasts [81].

Numerous papers using *Drosophila* as a model have examined potential connections between MCI, ROS, aging, and physiology. The consequences of MCI loss and the induction of ROS are context dependent. They are not always in obvious agreement, either from tissue to tissue or from study to study. For example, one analysis demonstrated that reverse electron transfer (RET) at MCI increases ROS production, and this occurs with the normal aging process [82]. Consistently, inhibition of RET decreases ROS and increases life span; this result would suggest that RET-induced ROS hurts survival [82]. However, a different study reported the opposite finding under stress-inducing conditions. For that scenario, inhibition of RET decreases ROS production and decreases life span—while activation of RET enhances ROS and extends life span. Those findings would suggest that ROS is protective under some conditions [83]. Effects are not limited to Complex I manipulation. *Drosophila* deficient for the Complex V subunit *bellweather* gene function demonstrate enhanced mitochondrial ROS; and downstream of this increased ROS, they have broadly enhanced levels of active zone material (BRP) in adult brains [84]. Genetically boosting mitochondrial function in adults has the opposite effect—attenuated BRP accumulation in the fly brain [84]. These findings closely match our larval motor neuron observations.

On the molecular level there are signaling puzzles uncovered by prior *Drosophila* work. For example, the unfolded protein response (UPR) is a key mediator when MCI function in lost. Neuronal impairment of the NDUFS1 (MCI subunit) homolog ND-75 elicits a spectrum of movement and seizure phenotypes, and it also activates the UPR on the cellular level [85]. All those *Drosophila* neuronal phenotypes can be reversed when yeast ND1 is exogenously added to the system, which restores NADH dehydrogenase function (while not restoring ATP production) [85]. Those findings suggest a damaging role in adult neurons for the MCI-deficient ROS-UPR axis. By contrast, another paper examined ND-75 disruption in adult muscle—and for this tissue, researchers found that elevated ROS levels trigger the UPR activation ultimately to preserve function [86]. That suggests a protective in in muscles for the ROS-UPR axis [86].

Our study expands upon these results and supports the idea that MCI has distinct roles at synaptic sites. Lack of MCI in larval motor neurons causes loss of mitochondria at synaptic terminals (Fig 1). This defect is linked to defective mitochondrial fusion (Fig 2), which is known to maintain mitochondrial integrity [87,88]. By contrast, lack of MCI in larval muscle appears to disassemble the postsynaptic density, which hurts neurotransmission (Figs 7–9).

**A novel form of presynaptic homeostatic plasticity triggered by ROS?**

ROS has been linked to short-term synaptic plasticity in *Drosophila* [53], as well as long-term potentiation (LTP) in mammals [89–91]. Here, we uncovered a role for ROS in augmentation of active zone material when MCI is impaired in neurons. This finding could be considered a form of homeostatic plasticity: an increase in active zone components likely drives potentiated release to compensate for defective baseline synaptic transmission, which would be expected after the loss of an energy source like mitochondria.

Homeostatic augmentations of active zone material have been reported at the *Drosophila* NMJ. For example, *rab3* mutants have an increase in active zone material to offset decreased synapse growth [92]. Additionally, ROS has been shown to be an obligate signal in *Drosophila* to maintain fundamental properties in both pre and postsynaptic compartments, including at the NMJ [53]. In our study, we observe an increase in active zone intensity (Fig 4), and this increase could overlap with mechanisms uncovered in those prior studies.

Alternatively, our results could be consistent with a form of homeostatic plasticity at the *Drosophila* NMJ called Presynaptic Homeostatic Potentiation (PHP). PHP is initiated when the activity of postsynaptic glutamate receptors is impaired. This decreases quantal size. The synapse detects the impairment, and muscle-to-nerve signaling drives an increase in presynaptic glutamate release [93,94]. This happens in part through an increase in influx of calcium into the neuron through $Ca_V2$ voltage-gated calcium channels [93,95–98]. PHP coincides with an increase in the size of the readily releasable pool (RRP) of synaptic vesicles [99–102] and apparent increases in active zone protein content or organization

[54,56,103]. The general model is that these modifications drive the neuron to release more glutamate, offsetting the initial synaptic challenge.

Could the excess mitochondrial ROS that is caused by the loss of Complex I be triggering the same or overlapping homeostatic mechanisms? It is possible. Quenching presynaptic ROS by expression of *Sod2* or by feeding flies NACA led to a reversion of active zone material back down to the control levels (Fig 4E). Under ideal conditions, control levels of active zone material would support normal neurotransmission (Fig 4A and 4J). However, when combined with presynaptic *NDUFS7* loss (already depleting synaptic mitochondria; Figs 1O and 2G), there is a loss in neurotransmission capacity (Fig 4M).

The molecular underpinnings of this mitochondrial-loss-induced homeostatic plasticity are unknown, but prior work offers ideas. One possibility is that *Sod2* changes the redox status of the entire cell, which could potentially affect active zone components [89]. Another possibility is that ROS could alter the release of calcium from intracellular stores such as ER; and in turn, this could induce calcium signaling to mitochondria, which would contribute to the formation of ATP and subsequent vesicle fusion [59,60]. Consistent with this idea, we observed a reduced synaptic strength at the terminals after simultaneous blockade of calcium release and import from ER stores to mitochondria through genetic or pharmacological manipulations (Fig 5). The link would not have to be a direct one. Indeed, a recent study indicated that activity-driven mitochondrial calcium uptake does not depend on the ER as a source of calcium to maintain normal synaptic strength [59]. Future studies can refine conclusions about presynaptic calcium dynamics downstream of MCI disruption. This is possible through reagents in the *Drosophila* neurogenetic toolkit like compart-targeted genetic-encoded calcium indicators [104].

## Loss of MCI in muscle and subsequent ROS accumulation diminish synaptic excitation

Little is known about what role postsynaptic ROS plays in regulating synaptic plasticity at the NMJ. It is unlikely that ROS is an abstract signal that triggers wholesale destruction. It likely has specific targets. There are clues from previous work. For example, ROS signaling plays direct roles in the activity-dependent structural plasticity of motor neurons and postsynaptic dendrites [53]. Additionally, postsynaptic ROS plays critical roles in dendritic development. In *Drosophila*, ROS appears to act as a developmental plasticity signal to regulate the size of the dendritic arbors. [105].

It is slightly surprising that the NMJ fails to compensate for a lack of MCI function in the muscle. This is because the NMJ employs multiple retrograde signals to stabilize function [106]. One hypothesis is that MCI deficiency and ROS interfere with important muscle-to-nerve signals normally required for maintaining NMJ setpoint function. Some of these signals are required to correct acute challenges (minutes) to NMJ function. In this context, some of the best characterized retrograde signals are: Bone Morphogenetic Protein (BMP) signaling [107–109]; the Insomniac (adaptor)-Cul3 (ubiquitin ligase)-Peflin (substrate) signaling complex [110]; and the endosomal recycling molecules class II PI3 kinase (PI3K) and Rab11 [111]. Additionally, the field has uncovered instructive signaling molecules from the muscle that maintain robust neurotransmission, including: muscle-secreted Semaphorin-2b [112]; and Target of Rapamycin (TOR) [113,114]. Future work can address whether these or similar targets are substrates of negative ROS regulation in the muscle.

The roles of ROS in muscle are less understood in terms of synapse regulation and function. In this study, we showed that excess ROS is sufficient to affect the organization of postsynaptic densities (Fig 7), ionotropic glutamate receptor clusters (Fig 8), and spectrin cytoskeleton (S11 Fig) in the muscle. The correlated uncoupling of pre and postsynaptic structures seems to be responsible for neurotransmission defects (Fig 9), and it likely coincides with broader structural instability [65,115]. These phenotypes are reminiscent of other *Drosophila* mutant phenotypes showing a degenerative NMJ (e.g., [65,115–117]). Consistent with a model of degeneration, we found behavioral defects in locomotion (Fig 9).

Despite the profound defects that result from muscle depletion of MCI, we were able to reverse many of those structural and functional problems by culturing animals in an antioxidant-rich media (0.5 mM NACA) or through muscle expression of *UAS-Sod2*. This is an interesting finding for future work because it points to mitochondrial ROS as a signal of

interest in the manifestation of synapse dysfunction. In subsequent work, we will take advantage of the *Drosophila* neuro-genetic toolkit to identify specific targets.

**Limitations and future directions: Linking mitochondrial ROS and synapse function**

There are fundamental questions that remain to be answered in future work. Many of these deal with ROS. First, we have not identified the specific ROS molecule(s) responsible for the synaptic phenotypes we have observed. Our data point to mitochondrial matrix ROS because *UAS-Sod2* overexpression ameliorates MCI loss phenotypes, while *UAS-Catalase* and *UAS-Sod1* do not. Additionally, *UAS-Sod2[RNAi]* phenocopies the ROS accumulation associated with MCI loss.

If the relevant ROS molecule is signaling from the mitochondrial matrix, this could mean superoxide ($O_2\bullet-$) or hydrogen peroxide ($H_2O_2$)—or less likely, another species, like hydroxyl radicals ($\bullet OH$). The function of superoxide dismutase enzymes is to convert superoxide into hydrogen peroxide [118]. Because SOD2 is mitochondrial and it has a mollifying effect on MCI dysfunction in our system, we would hypothesize that mitochondrial matrix superoxide ($O_2\bullet-$) is the likely culprit for the synaptic phenotypes we observe. This idea would explain why other antioxidants like SOD1, and Catalase do not help. SOD1 accumulates mainly in the cytoplasm [119,120], while Catalase accumulates mainly in peroxisomes [121,122]. NACA, which is the amide form of the antioxidant N-acetylcysteine, is membrane permeable; as such, it could accumulate in mitochondria [123]. We have not affirmatively demonstrated sub-compartment mitochondrial localization of NACA in our system. However, recent work by other groups is consistent with a reductive function of NACA for mitochondria [124,125].

Even though our data point to mitochondrial ROS, the mechanistic connection between matrix $O_2\bullet-$ and the synapse is not clear. Additionally, we do not know the relevant subcellular site(s) where ROS would need to accumulate to have its effects on the synapse. An intuitive idea is that the affected mitochondria and ROS would be near the synapse itself. For the NMJ, this is possible both in neurons and muscle. Yet our data do not neatly match that model. For example, six hours of rotenone exposure to the NMJ can enhance active zone accumulation, but only if the motor nerve is intact (S12 Fig). This finding suggests that the signaling downstream of mitochondrial ROS in neurons needs both time and space to work.

Concerning the muscle, mitochondrial ROS seems to have a damaging effect directly on the postsynaptic density. This effect hurts synaptic function. If the relevant site of mitochondrial ROS in muscle happens to be closer to the NMJ than the relevant site in neurons, then the damaging effects from the muscle could partly a function of proximity. Determining if it matters whether mitochondrial ROS is physically near the NMJ could inform the possible signaling modalities that govern mitochondrial regulation of synapse function.

## Resource availability

### Lead contact

The lead contact for this study is Dr. C. Andrew Frank (andy-frank@uiowa.edu).

### Materials availability

Dr. C. Andrew Frank's laboratory uses the fruit fly *Drosophila melanogaster* as a model organism. Consistent with ethical conduct of research, the lab maintains *Drosophila* stocks that it generates and publishes, as well as useful precursors and derivatives of those stocks. These stocks are available to researchers who make requests to Dr. Frank (andy-frank@uiowa.edu)—or to the appropriate original source—and they are shipped promptly.

### Data tables

Summary data are in Tables A–U in the S1 Tables file. Raw data are housed in Excel file tables. The S1 Data Excel file contains raw data for the main figures. The S2 Data Excel file contains raw data for the supplemental figures.

## STAR methods

Please see Table V in the S1 Tables file for detailed information about the reagents and materials described below.

### Experimental model details

**Drosophila husbandry.** *Drosophila melanogaster* was cultured on a standard cornmeal media containing molasses and yeast prepared according to the Bloomington *Drosophila* Stock Center (BDSC, Bloomington, IN, USA) recipe. Fruit fly husbandry was performed according to standard practices [126]. Both male and female larvae were used for this study. Larvae were raised at 25 °C in humidity-controlled and light-controlled Percival DR-36VL incubators (Geneva Scientific). For drug feeding experiments in larvae, the control, RNAi, and mutant animals were grown throughout life (egg laying to analysis) in media containing the relevant drug. Those drugs were 0.5 mM curcumin (Sigma Aldrich), 0.5 mM N-acetyl cysteine amide (NACA) (Sigma Aldrich), or 50 µM rotenone (Sigma Aldrich).

**Drosophila stocks.** $w^{1118}$ was used as a nontransgenic wild-type stock [127]. The GAL4 drivers used in this study were $elaV^{C155}$-Gal4 [128], BG57-Gal4 [64], and D42-Gal4 [129]. For the neuronal experiments, a switch from *D42-Gal4* (motor neuron) to $elaV^{C155}$-Gal4 (pan-neuronal) was made between Figs 4 and 5 for practicality of genetic stock building with several markers in the latter experiments. The core neuronal results remained unchanged.

Several *UAS-RNAi* and genetic mutant lines were obtained from the Bloomington *Drosophila* Stock Center or directly from researchers who produced them (Table V in S1 Tables includes the Bloomington Line (BL) numbers). We verified the effectiveness of *UAS-ND-20L[RNAi]* ($ND-20L^{HMJ23777}$, now termed *UAS-NDUFS7[RNAi]*) in this study. Other mutant and transgenic lines have been characterized in prior studies. These include: $UAS-IP_3$-sponge ($UAS-IP_3$-Sponge.m30 shared by Dr. Masayuki Koganezawa's lab) [126,130]; *UAS-Catalase* [131,132]; *UAS-Sod1* [133]; *UAS-Sod2* [47]; *UAS-Sod2 [RNAi]* [134]; *Df(3L)ED4288* [135]; $ND-30^{epgy}$ [136]; *UAS-ND-30[RNAi]* [12]; *UAS-mitoGFP* [137]; *UAS-marf[RNAi]* [31]; *UAS-drp1[RNAi]* [138]; *UAS-CalX[RNAi]* [139]; *UAS-mcu[RNAi]* [140,141]; *UAS-hexokinase A[RNAi]* [142]; *UAS-hexokinase C[RNAi]* [142]; *UAS-idh[RNAi]* [143]; *UAS-Cit.(Si)-Syn[RNAi]* [144]; *UAS-Scsα[RNAi]* [145].

### Method details

**Immunohistochemistry.** Wondering third instar larvae were dissected and fixed on a sylgard Petri plate in ice-cold HL3 and fixed in 4% paraformaldehyde in PBS for 30 min or in Bouin's fixative for 2 min as described earlier [146]. Briefly, larvae were washed with PBS containing 0.2% Triton X-100 (PBST) for 30 min, blocked for an hour with 5% normal goat serum in PBST, and incubated overnight in primary antibodies at 4 °C followed by washes and incubation in secondary antibodies. Monoclonal antibodies: anti-Dlg (4F3), anti-DGluRIIA (8B4D2), anti-CSP (ab49), anti-Synapsin (3C11), anti-Futsch (22C10), anti-Bruchpilot (nC82), and anti-α-Spectrin (3A9) were obtained from the Developmental Studies Hybridoma Bank (University of Iowa, IA, USA) and were used at 1:30 dilution. Rabbit anti-GFP (Abcam) was used at 1:200 dilutions. Anti-GluRIII (1:100) ([147]; Table V in S1 Tables) was gifted by Aaron DiAntonio (Washington University, St. Louis, USA). Fluorophore-coupled secondary antibodies Alexa Fluor 488, Alexa Fluor 568, or Alexa Fluor 647 (Molecular Probes, ThermoFisher Scientific) were used at 1:400 dilution. Alexa 488 or 647 and Rhodamine conjugated anti-HRP were used at 1:800 and 1:600 dilutions, respectively. The larval preparations were mounted in VECTASHIELD (Vector Laboratories, USA) and imaged with a laser scanning confocal microscope (LSM 710; Carl Zeiss). All the images were processed with Adobe Photoshop 7.0 (Adobe Systems, San Jose, CA, USA).

**Confocal imaging, quantification, and morphometric analysis.** Samples were imaged using a Carl Zeiss scanning confocal microscope equipped with 63×/1.4 NA oil immersion objective using separate channels with four laser lines (405, 488, 561, and 637 nm) at room temperature. The stained NMJ boutons were counted using anti-Synapsin or anti-HRP co-stained with anti-Dlg on muscle 6/7 of A2 hemisegment, considering each Synapsin or HRP punctum to be a bouton. At least 8 NMJs were used for bouton number quantification. For fluorescence quantifications of GluRs, Dlg, BRP,

α-Spectrin, HRP, and CSP, all genotypes were immunostained in the same tube with identical reagents, then mounted and imaged in the same session. Z-stacks were obtained using identical settings for all genotypes with z-axis spacing between 0.2 and 0.5 μm and optimized for detection without saturation of the signal.

**ROS and rotenone incubation assay.** Larvae were dissected in ice-cold calcium-free HL3 to label ROS in neurons. ROS levels were detected in mitochondria by incubating live preparation in 1× Schneider's media with MitoSOX Red (Molecular Probes, ThermoFisher Scientific) fluorogenic dye at 1:200 dilutions for 20–30 min. Briefly, larvae were washed with HL3, mounted in VECTASHIELD (Vector Laboratories, USA), and immediately imaged in a laser scanning confocal microscope (LSM 710; Carl Zeiss). To study the effect of DMSO and rotenone (Sigma Aldrich) in BRP, larvae were dissected in HL3 and incubated in 1× Schneider's media containing either 500 μM of rotenone or DMSO. After every 30 min, the old media was replaced with fresh media containing rotenone or DMSO. The above preparations were fixed with 4% paraformaldehyde, stained with anti-nc82 antibodies, mounted and imaged in a confocal microscope.

**Electrophysiology and pharmacology.** All dissections and recordings were performed in modified HL3 saline [148] containing 70 mM NaCl, 5 mM KCl, 10 mM MgCl$_2$, 10 mM NaHCO$_3$, 115 mM sucrose, 4.2 mM trehalose, 5 mM HEPES, and 0.5 mM CaCl$_2$ (unless otherwise noted), pH 7.2. Neuromuscular junction sharp electrode (electrode resistance between 20 and 30 MΩ) recordings were performed on muscles 6/7 of abdominal segments A2 and A3 in wandering third-instar larvae as described [126]. Recordings were performed on a Leica microscope using a 10× objective and acquired using an Axoclamp 900A amplifier, Digidata 1440A acquisition system, and pClamp 10.7 software (Molecular Devices). Electrophysiological sweeps were digitized at 10 kHz and filtered at 1 kHz. Data were analyzed using Clampfit (Molecular Devices) and MiniAnalysis (Synaptosoft) software. Miniature excitatory postsynaptic potentials (mEPSPs) were recorded in the absence of any stimulation and motor axons were stimulated to elicit excitatory postsynaptic potentials (EPSPs).

**Larval crawling assay.** Vials containing third instar larvae were analyzed for this assay. Vials were poured with 4 ml of 20% sucrose solution and left for 10 min to let the larvae float on top. Floating third instar animals were poured into a petri dish and washed gently twice with deionized water in a paintbrush. A minimum of 10 larvae of each genotype were analyzed on a 2% agarose gel in a petri dish with gridline markings 1 cm on graph paper. The larvae were acclimatized in the petri dish before videotaping. The average distance crawled (in centimeters) by larvae was calculated based on the average number of gridlines passed in 30 s [146].

**Real-time PCR (RT-PCR).** Total RNA was extracted from 10 larval muscle fillets or 20 larval brains for each experiment using TRIzol (GIBCO) according to the manufacturer's instructions. Concentration and purity of total RNA were measured using a NanoDrop. 1 μg of total RNA was then subjected to DNA digestion using DNase I (Ambion). This was immediately followed by reverse transcription using the iScript Reverse Transcription Supermix (1708841, Bio-Rad). qPCR was performed using the StepOnePlus instrument (ThermoFisher Scientific) and SYBR Green Supermix (172-5270, Bio-Rad) by following the manufacturer's instructions. The Step One Software analyzed the qPCR, and the relative expression level was presented as the ratio of the target gene to the internal standard gene, *rp49*. Each sample was analyzed in triplicate. At least three independent biological repeats were obtained for each experiment. The following primers were used for quantitative PCR (qPCR) analysis. For the *ND-20L* gene, forward: 5′ CATGCCGGTGTACGATTACC 3′ and reverse: 5′ CGTCCCCAGTTTAGCAGGTC 3′ primers are used. Similarly, for analyzing the *ND-20* transcript, forward: 5′ GAAGTGGCCCAAAATCTGCC 3′ and reverse: 5′ GAGCAGATCGTCCAGTCTGG 3′ primers were used, respectively. All data for the RT-PCR are presented as mean ± standard deviation.

**Seahorse analysis for OCR measurement.** The Seahorse analysis was carried out as previously described [149] with some modifications. Briefly, mitochondria were isolated from the thoracic muscles of 20 adult *Drosophila melanogaster* using a differential centrifugation method. Thoraces were homogenized in cold isolation buffer, followed by a 300×*g* spin to remove debris and a subsequent 3,000×*g* spin to pellet mitochondria. The final mitochondrial pellet was resuspended in mitochondrial isolation buffer, and protein concentration was quantified using a BCA assay. Isolated mitochondria from *Drosophila melanogaster* thoracic muscles were resuspended in Agilent Seahorse XF assay medium

supplemented with glucose (10 mM), pyruvate (1 mM), and glutamine (2 mM), pH 7.4. For the Seahorse assay, ~5–10 µg of isolated mitochondria were added to each well of a Seahorse XF Cell Culture Microplate. After centrifugation (3,000×$g$ for 20 min at 4 °C), the plate was then loaded into the Seahorse XFeMini Analyzer, and the mitochondrial respiration profile was measured. The OCR data were normalized to protein concentration. All data for the Seahorse assay are presented as mean ± standard deviation.

**Rendering schematic cartoons, figures, and tables.** For this study, the schematic cartoons for Fig 4 were drawn for this study using BioRender software. The schematic cartoons for Fig 5 were drawn using Adobe Illustrator. All final Figures or Supplemental Figures were saved as TIF files and sized in Adobe Photoshop. Tables were generated in Microsoft Word.

## Quantification and statistical analysis

**Quantification of mitochondrial branch length.** Controls and RNAi-depleted animals were analyzed using mitochondrial marker Mito-GFP, *D42* driver in cell bodies from larvae VNC. All images were processed using ImageJ software. Z-stacks of individual neurons were merged. The Mito-GFP signal was enhanced by adjusting brightness and contrast. The binary masks were created using Image>Adjust>Threshold, Method:Otsu, and Background:Dark. The branches were generated using Process>Binary>Skeletonized [150]. These skeletonized images were analyzed, and branch length of the individual cluster was manually calculated using ImageJ tools.

**Imaging quantifications.** Maximum intensity projections were used for quantitative image analysis with the ImageJ software (National Institutes of Health) analysis toolkit. Boutons from muscle 4 or type Ib terminal boutons on the muscle 6/7 of A2 hemisegment from at least six NMJ synapses were used for quantification using Image J software. Student $t$ test for pairwise and one-way ANOVA with post-hoc Tukey's test for multiple comparisons was used for statistical analysis, using GraphPad Prism Software. Specific $p$-value and tests are noted in the figures and figure legends and supplemental files and shown in graphs as follows: * $p < 0.05$, ** $p < 0.001$, and *** $p < 0.0001$. The data are presented as mean ± s.e.m. For quantification of Futsch loops, third instar larval preparations were double immunostained with HRP, 22C10 and images were captured in Zeiss LSM710 confocal microscope. Only NMJs of muscles 6/7 of A2 hemisegment were used for quantification. The images were digitally magnified using ImageJ software and the total number of HRP-positive boutons was manually counted in each image. Futsch positive loops, which are co-localized with HRP, were included in this analysis. Images with incomplete loops and diffused staining were not included in the count [151].

**Electrophysiological analysis.** Average mEPSP, EPSP, and QC were calculated for each genotype by dividing EPSP amplitude by mEPSP amplitude. Muscle input resistance ($R_{in}$) and resting membrane potential ($V_{rest}$) were monitored during each experiment. Recordings were rejected if the $V_{rest}$ was above −60 mV, and $R_{in}$ was less than 5 MΩ. Pharmacological agents were bath applied in recording saline at the final concentrations indicated in the text, figures, and tables. The agents included Xestospongin C (Abcam), Dantrolene (Tocris), and BAPTA-AM (Sigma Aldrich). Failure analysis was performed in HL3 solution containing 0.1 mM $CaCl_2$, which resulted in failures in about half of the stimulated responses in wild-type larvae. A total of 30 trials (stimulations) were performed at each NMJ in all genotypes. The failure rate was obtained by dividing the total number of failures by the total number of trials (100). High-frequency (10 Hz) recordings were performed at a calcium concentration of 2 mM and paired-pulse recordings (10 Hz) were performed at calcium concentrations of 0.4 mM and 1.5 mM, respectively. Paired-pulse ratios were calculated as the EPSP amplitude of the second response divided by the first response (EPSP2/EPSP1).

## Supporting information

**S1 Fig. Related to Fig 1. Tissue-specific expression of the Mitochondrial Complex I subunit NDUFS7 and functional assessment of mitochondrial respiration via Seahorse-based oxygen consumption rate (OCR) analysis.**

(**A** and **B**) Quantitative RT-PCR showing transcript levels of *ND-20L* in Gal4 controls and pan-neuronal and muscle Gal4-driven *UAS-NDUFS7[RNAi]*. Compared to pan-neuronal Gal4 control (*elaV(C155)-Gal4/+*), *elaV(C155)-Gal4*-driven *UAS-NDUFS7[RNAi]* (*elaV(C155)-Gal4/+; NDUFS7[RNAi]/+*) led to ~25% reduction in *ND-20L* transcript level in neurons. The muscle Gal4-driven *UAS-NDUFS7[RNAi]* showed ~50% reduction in *ND-20L* transcript levels (*UAS-NDUFS7[RNAi]/+; BG57-Gal4/+*) compared to Gal4 control (*BG57-Gal4/+*) in the muscle. Error bars represent mean ± standard deviation $p = 0.354$, $*p = 0.015$. Statistical analysis based on Student $t$ test for pairwise comparisons. (**C** and **D**) Quantitative RT-PCR showing transcript levels of *ND-20* in Gal4 controls and pan-neuronal and muscle Gal4-driven *UAS-NDUFS7[RNAi]*. Compared to pan-neuronal Gal4 control (*elaV(C155)-Gal4/+*), *elaV(C155)-Gal4*-driven *UAS-NDUFS7[RNAi]* (*elaV(C155)-Gal4/+; UAS-NDUFS7[RNAi]/+*) led to ~70% reduction in *ND-20* transcript level in neurons. The muscle Gal4-driven *UAS-NDUFS7[RNAi]* showed ~60% reduction in *ND-20* transcript levels (*UAS-NDUFS7[RNAi]/+; BG57-Gal4/+*) compared to Gal4 control (*BG57-Gal4/+*) in the muscle. Error bars represent mean ± standard deviation $***p = 0.001$, $***p = 0.0003$. Statistical analysis based on Student's $t$ test for pairwise comparisons. (**E**) Histogram showing OCR in the mitochondria isolated from the thoracic region in muscle Gal4 control and muscle-depleted *UAS-NDUFS7[RNAi]* animals. The muscle depletion of *NDUFS7* message resulted in a ~80% reduction in oxygen consumption compared to the Gal4 control. (**F**) OCR in muscle Gal4 control and muscle-depleted *UAS-NDUFS7[RNAi]* are plotted on a 60-min time scale. Raw data for this figure are available in the S2 Data Excel file, tab S1 Fig.
(TIF)

**S2 Fig. Related to Fig 2. Loss of an MCI subunit triggers excess mitochondrial ROS accumulation in neurons and muscle.** (**A–D**) Representative confocal images of the ventral nerve cord (VNC) at the third instar larval brain in (A) *mito-GFP, D42-Gal4/+*, (B) *UAS-NDUFS7[RNAi]/+; mito-GFP, D42-Gal4/+*, (C) *UAS-NDUFS7[RNAi]/+; mito-GFP, D42-Gal4/+* with NACA, and (D) *UAS-NDUFS7[RNAi]/UAS-Sod2; mito-GFP, D42-Gal4/+* labeled with superoxide indicator MitoSOX (magenta) in live animals. *UAS-NDUFS7[RNAi]*-depleted larvae supplemented with NACA or co-expressing *Sod2* in neurons showed a significant correction in mitochondrial ROS levels compared to the *NDUFS7[RNAi]* knock-down tissues. (**E–H**) Confocal images of third instar larval body wall muscle 6 at A2 hemisegment in the indicated genotypes. Muscles depleted of MCI function by *UAS-NDUFS7[RNAi]* had ROS corrected back down to normal levels when the animals were supplemented with NACA or when there was co-expression of a *UAS-Sod2* transgene. Scale bar: 5 μm. (**I** and **J**) Histogram showing the relative intensity of mitoSOX in VNC and body wall muscle in (E) *UAS-mitoGFP/+; BG-57-Gal4/+*, (F) *UAS-NDUFS7[RNAi]/UAS-mitoGFP; BG57-Gal4/+*, (G) *UAS-NDUFS7[RNAi]/UAS-mitoGFP; BG-57-Gal4/+*, and (H) *UAS-NDUFS7[RNAi]/UAS-Sod2; BG57-Gal4/+* labeled with superoxide indicator MitoSOX (magenta) in live animals. $***p < 0.0001$; ns, not significant. Statistical analysis based on one-way ANOVA followed by post-hoc Tukey's multiple-comparison test. Error bars represent mean ± s.e.m. Raw data for this figure are available in the S2 Data Excel file, tab S2 Fig.
(TIF)

**S3 Fig. Related to Fig 2. Loss of *Sod2* induces excessive ROS formation in neuronal and muscle tissues.** (**A, -A′** and **B, B′**) Representative confocal images of the ventral nerve cord (VNC) at the third instar larval brain in the genotypes (A, A′) *UAS-mito-GFP, D42-Gal4/+* and (B, B′) *UAS-mito-GFP, D42-Gal4/Sod2[RNAi]* labeled with superoxide indicator MitoSOX (magenta) and mito-GFP (green) in live tissue. (**C**) Histogram showing the relative intensity of MitoSOX in the VNC of indicated genotypes. $***p = 0.0003$ (VNC: control versus *Sod2[RNAi]*). (**D, D′** and **E, E′**) Representative confocal images of the axon at the third instar larval fillet in (D, D′) UAS-*mito-GFP, D42-Gal4/+* and (E, E′) *UAS-mito-GFP, D42-Gal4/UAS-Sod2[RNAi]* labeled with superoxide indicator MitoSOX (magenta) and mitoGFP (green) in live tissue. (**F**) Histogram showing the relative intensity of MitoSOX in axons in the indicated genotypes. $*p = 0.020$ (Axon: control versus *Sod2* RNAi). (**G, G′** and **H, H′**) Representative confocal images of a third instar bouton of synapse 6/7 in the A2 hemisegment in (G, G′) *UAS-mito-GFP, D42-Gal4/+*, and (H, H′) *UAS-mito-GFP, D42-Gal4/Sod2[RNAi]* labeled with superoxide indicator

MitoSOX (magenta) and mitoGFP (green) in live tissue. (**I**) Histogram showing the relative intensity of MitoSOX at boutons in the indicated genotypes. ***$p < 0.0001$ (boutons: control versus Sod2 RNAi). (**J, J′** and **K, K′**) Representative confocal images of the third instar 6/7 muscle of A2 hemi segment in (J-J′) *UAS-mito-GFP/+; BG57-Gal4/+*, and (**K, K′**) *UAS-mito-GFP/+; BG57-Gal4/Sod2[RNAi]* labeled with superoxide indicator MitoSOX (magenta) and mitoGFP (green) in live animals. (**L**) Histogram showing the relative intensity of mitoSOX in 6/7 muscle in the indicated genotypes. ***$p = 0.0001$ (6/7 muscles: control versus *Sod2 [RNAi]*). The depletion of *Sod2* by RNAi results in the abnormal accumulation of reactive oxygen species (ROS) in neurons and muscles. Scale bar: 10 μm (A, A′–E, E′) and 5 μm (G, G′ and K, K′). Statistical analysis based on Student's *t* test for pairwise comparison. Error bars represent mean ± s.e.m. Raw data for this figure are available in the S2 Data Excel file, tab S3 Fig.
(TIF)

**S4 Fig.  Related to Fig 2. Misregulation of ROS in neurons affects mitochondrial morphology and distribution in the ventral nerve cord and distal axons in third instar larvae.** *UAS-NDUFS7[RNAi], UAS-NDUFS7[RNAi]* co-expressing *UAS-Sod2, ND-30[RNAi]*, and *ND-30[RNAi]* co-expressing *UAS-Sod2* and controls were crossed to a motor neuron driver line (*D42-Gal4, UAS-mitoGFP*) to label neuronal mitochondria. (**A**) Ventral nerve cord (VNC): *UAS-mitoGFP* exhibits normal mitochondrial organization, *UAS-NDUFS7[RNAi]* and *ND-30 RNAi* exhibit clustered mitochondria. However, when *UAS-NDUFS7[RNAi]* animals were raised in a media containing NACA or when there was co-expression of *UAS-Sod2* in *UAS-NDUFS7[RNAi]*- and *ND-30[RNAi]*-depleted animals, there was a restoration of normal mitochondrial organization. The respective fluorescent images were skeletonized to measure mitochondrial branch length (organization). (**B**) Comparison of a proximal axonal segment in A2 and a distal segment in A5. Distal segments of A5 axons in *UAS-NDUFS7[RNAi]* contain fewer mitochondria than proximal segments. Scale bar: 5 μm. Mitochondrial distribution was significantly suppressed back to control values when *UAS-NDUFS7[RNAi]* was raised in media containing NACA or genetically over-expressing *UAS-Sod2* in the *UAS-NDUFS7[RNAi]* and *ND-30[RNAi]* backgrounds in neurons. (**C** and **D**) Histogram showing mitochondrial branch length (μm) and number in the indicated genotypes. ***$p < 0.0001$, ns, not significant. Statistical analysis based on one-way ANOVA followed by post-hoc Tukey's multiple-comparison test. Error bars represent mean ± s.e.m. Raw data for this figure are available in the S2 Data Excel file, tab S4 Fig.
(TIF)

**S5 Fig.  Related to Fig 2. Neither *UAS-Catalase* nor *UAS-Sod1* overexpression rescue the synaptic and mitochondrial phenotypes in *NDUFS7*-depleted animals.** Mitochondrial morphology and distribution are affected in the ventral nerve cord and distal axons. *Drosophila* stocks of *UAS-NDUFS7[RNAi], UAS-NDUFS7[RNAi]* co-expressing *UAS-Catalase* or *UAS-Sod1*, as well as controls,were crossed to a motor neuron driver (*D42-Gal4, UAS-mitoGFP*) to label neuronal mitochondria. (**A**) Ventral nerve cord (VNC): *UAS-mitoGFP, UAS-Cat*, and *UAS-Sod1* exhibit normal mitochondrial organization. By contrast, *NDUFS7[RNAi]* exhibits clustered mitochondria. Co-expression of *UAS-Cat* or *UAS-Sod1* in the *NDUFS7[RNAi]* background does not restore normal mitochondrial organization. The respective fluorescent images were skeletonized to measure mitochondrial branch length (organization). Scale bar: 10 μm. (**B**) Comparison of a proximal axonal segment in A2 and a distal segment in A5. Distal segments of A5 axons in *NDUFS7[RNAi]* contain fewer mitochondria than proximal segments. Scale bar: 5 μm. Mitochondrial distribution was not restored to control values when *NDUFS7[RNAi]* was coexpressed with *UAS-Cat* or *UAS-Sod1* in neurons. (**C** and **D**) Histogram showing mitochondrial branch length (μm) and number in the indicated genotypes. ***$p < 0.0001$, ns, not significant. Statistical analysis based on one-way ANOVA followed by post-hoc Tukey's multiple-comparison test. Error bars represent mean ± s.e.m.
(TIF)

**S6 Fig.  Related to Fig 2. Depleting a mitochondrial fusion gene induces excess mitochondrial ROS formation in neurons.** (**A, A′–D, D′**) Representative confocal images of the ventral nerve cord (VNC) at the third instar larval brain in (A, A′) *mito-GFP, D42-Gal4/+*, (B, B′) *mito-GFP, D42-Gal4/marf[RNAi]*, (C, C′) *mito-GFP, D42-Gal4/+* with 25 μM rotenone

(ROT), and (D, D′) *UAS-NDUFS7[RNAi]/+; mito-GFP, D42-Gal4/+* labeled with superoxide indicator MitoSOX (magenta) and mitoGFP (green) in live animals. (**E, E′–H, H′**) Representative confocal images of the axon at the third instar larval fillet in (E, E′) *mito-GFP, D42-Gal4/+*, (F, F′) *mito-GFP, D42-Gal4/marf[RNAi]*, (G, G′) *mito-GFP, D42-Gal4/+* with 25 μM ROT and (H, H′) *UAS-NDUFS7[RNAi]/+; mito-GFP, D42-Gal4/+* labeled with superoxide indicator MitoSOX (magenta) and mitoGFP (green) in live animals. (**I, I′–L, L′**). Representative confocal images of the third instar bouton of A2 hemisegment in (II') *mito-GFP, D42-Gal4/+*, (J, J′) *mito-GFP, D42-Gal4/marf[RNAi]*, (K, K′) *mito-GFP, D42-Gal4/+* with 25 μM ROT and (L-L') *UAS-NDUFS7[RNAi]/+; mito-GFP, D42-Gal4/+* labeled with superoxide indicator MitoSOX (magenta) and mitoGFP (green) in live animals. The depletion of *marf* by RNAi or blocking MCI activity induces abnormal accumulation of ROS in neurons. Scale bar: 10 μm. (M-O) Histogram showing the relative intensity of MitoSOX in VNC, axon and boutons in the indicated genotypes ***$p < 0.0001$; **$p < 0.001$; **$p = 0.002$ (Bouton: control versus *marf[RNAi]*), *$p < 0.05$. Statistical analysis based on one-way ANOVA followed by post-hoc Tukey's multiple-comparison test. Error bars represent mean ± s.e.m. Raw data for this figure are available in the S2 Data Excel file, tab S6 Fig.
(TIF)

**S7 Fig. Related to Fig 2. Loss of an MCI subunit in motor neurons causes abnormal Cysteine String Protein (CSP) accumulation in distal axons.** Representative confocal images of the proximal (A2) and distal (A5) axons of larvae in (**A, A′** and **B, B′**) Gal4 control, (**C, C′** and **D, D′**) *UAS-Sod2/+; D42-Gal4/+*, (**E, E′** and **F, F′**) *UAS-NDUFS7[RNAi]/+; D42-Gal4/+*, (**G, G′** and **H, H′**) *UAS-NDUFS7[RNAi]/+; D42-Gal4/+* with NACA, and (**I, I′** and **J, J′**) *UAS-NDUFS7[RNAi]/ UAS-Sod2; D42-Gal4/+*. Axons were double immunolabeled with CSP (magenta) and HRP (green) antibodies. Motor neuron-depleted *UAS-NDUFS7[RNAi]* larval axons showed abnormal accumulation of CSP at a more distal hemisegment (A5) compared to Gal4 controls. Scale bar: 10 μm. CSP aggregates were cleared when *UAS-NDUFS7[RNAi]* animals were raised in media containing NACA or genetically expressing *UAS-Sod2* in neurons. (**K**) Schematic illustration showing VNC, axons and body wall muscle in a third instar larvae (**L**) Histogram showing the percentage of axons with abnormal accumulation in the indicated genotypes. ***$p = 0.0008$ (control: normal versus accumulation), *$p = 0.0008$ (*UAS-NDUFS7[RNAi]*: normal versus accumulation), *$p = 0.0179$ (*Sod2* OE: normal versus accumulation), *$p = 0.0179$ (*UAS-NDUFS7[RNAi]* with NACA: normal versus accumulation), *$p = 0.0004$ (*UAS-NDUFS7[RNAi]* with *Sod2* OE: normal versus accumulation), **$p = 0.0046$ (*UAS-NDUFS7[RNAi]* versus *UAS-NDUFS7[RNAi]* with NACA: accumulation) and ***$p = 0.0006$ (*UAS-NDUFS7[RNAi]* versus *UAS-NDUFS7[RNAi]* with *Sod2* OE: accumulation). Statistical analysis was based on Fisher's exact test to differentiate two distinct phenotypes in the same sample. Error bars represent mean ± s.e.m. Raw data for this figure are available in the S2 Data Excel file, tab S7 Fig.
(TIF)

**S8 Fig. Related to Fig 2. Neither *Catalase* nor *Sod1* overexpression resolve Cysteine String Protein (CSP) accumulation in distal axons of *NDUFS7* RNAi-expressing animals.** Representative confocal images of the proximal (A2) and distal (A5) axons of larvae in (**A, A′, B, B′**) Gal4 control: *UAS-mitoGFP, D42-Gal4*, (**C, C′, D, D′**) *NDUFS7[RNAi]/+; UAS-mitoGFP, D42-Gal4/+*, (**E, E′, F, F′**) *UAS-Cat/+; UAS-mitoGFP, D42-Gal4/+*, (**G, G′, H, H′**) *UAS-Sod1/+; UAS-mitoGFP, D42-Gal4/+*, (**I, I′, J, J′**) *NDUFS7[RNAi]/UAS-Cat; UAS-mitoGFP, D42-Gal4/+*, and (**J, J′, K, K′**) *NDUFS7[RNAi]/UAS-Sod1; UAS-mitoGFP, D42-Gal4/+*. Axons were double immunolabeled with CSP (magenta) and HRP (green) antibodies. Motor neuron-depleted *NDUFS7[RNAi]* larval axons showed abnormal accumulation of CSP in axons at the more distal hemisegment (A5) compared to Gal4 controls. Scale bar: 10 μm. CSP aggregates were not cleared when *NDUFS7[RNAi]* animals were concurrently expressing *UAS-Cat* or *UAS-Sod1* in neurons. (**M**) Histogram showing the percentage of axons with abnormal accumulation in the indicated genotypes. ***$p < 0.0001$ (control: normal vs. accumulation), ***$p < 0.0001$ (*NDUFS7[RNAi]*: normal vs. accumulation), ***$p < 0.0001$ (*Cat* OE: normal vs. accumulation), ***$p < 0.0001$ (*Sod1* OE: normal vs. accumulation), ***$p < 0.0001$ (*NDUFS7[RNAi]+Cat* OE: normal vs. accumulation), ***$p < 0.0001$ (*NDUFS7[RNAi]+Sod1* OE: normal vs. accumulation), $p = 0.911$ (*NDUFS7[RNAi]* vs. *NDUFS7[RNAi]+Cat*

OE: accumulation) and $p = 1.00$ (*NDUFS7[RNAi]* vs. *NDUFS7[RNAi]+Sod1* OE: accumulation). Statistical analysis was based on Fisher's exact test to differentiate two distinct phenotypes in the same sample. Error bars represent mean ± s.e.m. Raw data for this figure are available in the S2 Data Excel file, tab S8 Fig.
(TIF)

**S9 Fig. Related to Fig 3. Neither *Catalase* nor *Sod1* overexpression restore cytoskeletal loops to *NDUFS7-* deficient animals.** Representative confocal images of NMJ synapses at muscle 6/7 of (**A, A′**) *UAS-mitoGFP, D42-Gal4* control, (**B, B′**) *UAS-Cat* overexpression (*UAS-Cat/+; UAS-mitoGFP, D42-Gal4/+*), (**C, C′**) *UAS-Sod1* overexpression (*UAS-Sod1/+; UAS-mitoGFP, D42-Gal4/+*), (**D, D′**) *UAS-mitoGFP, D42-Gal4*-driven *NDUFS7[RNAi]* (*UAS-NDUFS7 [RNAi]/+; UAS-mitoGFP, D42-Gal4/+*), (**E, E′**) *NDUFS7* knockdown with *UAS-Cat* (*UAS-NDUFS7[RNAi]/UAS-Cat; UAS-mitoGFP, D42-Gal4/+*) and (**F, F′**) *NDUFS7* knockdown with *UAS-Sod1* (*UAS-NDUFS7[RNAi]/UAS-Sod1; UAS-mitoGFP, D42-Gal4/+*). Each condition was double immunolabeled with 22C10 (anti-Futsch, magenta) and anti-HRP (green) antibodies. The motor neuron-depleted *NDUFS7[RNAi]* larvae showed a decrease in the number of Futsch-positive loops as compared to the Gal4 control. Neither *UAS-Cat* nor *UAS-Sod1* restored Futsch-positive loops in the *UAS-NDUFS7[RNAi]* background. Scale bar: 5 μm. (**G**) Histograms showing the percentage of Futsch-positive loops in the indicated genotypes. $p = 0.659$ (Control versus *UAS-Cat*), $p = 0.505$ (Control versus *UAS-Sod1*), $p < 0.0001$ (Control versus *UAS-NDUFS7[RNAi]*), $p = 0.071$ (*UAS-NDUFS7[RNAi]* versus *UAS-NDUFS7[RNAi] + UAS-Cat*) and $p = 0.039$ (*UAS-NDUFS7[RNAi]* versus *UAS-NDUFS7[RNAi] + UAS-Sod1*). Statistical analysis based on one-way ANOVA followed by post-hoc Tukey's multiple-comparison test. Error bars represent mean ± s.e.m. Raw data for this figure are available in the S2 Data Excel file, tab S9 Fig.
(TIF)

**S10 Fig. Related to Fig 2. The Mitofusin gene *marf* is required presynaptically to maintain NMJ synaptic transmission.** (**A–F**) Representative electrophysiological traces of mEPSPs and EPSPs in (**A**) motor-neuron Gal4 control *D42-Gal4/+*, (**B**) *UAS-NDUFS7[RNAi]/+; D42-Gal4/+*, (**C**) *UAS-marf[RNAi]/D42-Gal4*, (**D**) *UAS-Sod2 OE/+; UAS-marf[RNAi]/ D42-Gal4*, (**E**) *UAS-NDUFS7[RNAi]/+; UAS-marf[RNAi]/D42-Gal4*, and (**F**) *UAS-NDUFS7[RNAi]/UAS-Sod2; UAS-marf [RNAi]/D42-Gal4*. Scale bars for EPSPs (mEPSP) are $x = 50$ ms (1,000 ms) and $y = 10$ mV (1 mV). Note that mEPSP and EPSP amplitudes were reduced in motor neuron Gal4-driven *UAS-marf[RNAi]* and *UAS-marf[RNAi] + UAS-NDUFS7 [RNAi]* animals. (**G–I**) Histograms showing average mEPSPs, EPSPs, and quantal content in the indicated genotypes. A minimum of 8 NMJ recordings of each genotype were used for quantification. *$p < 0.05$ (mEPSP amplitude: control vs. *UAS-NDUFS7[RNAi]*), *$p = 0.007$ (QC: control vs. *UAS-marf[RNAi]*),*$p = 0.026$ (QC: *UAS-marf[RNAi]* vs. *UAS-marf[RNAi] + Sod2 OE*), **$p < 0.001$, ***$p < 0.0001$, ns, not significant. Statistical analysis based on one-way ANOVA followed by post-hoc Tukey's multiple-comparison test. Error bars represent mean ± s.e.m. Raw data for this figure are available in the S2 Data Excel file, tab S10 Fig.
(TIF)

**S11 Fig. Related to Fig 4. *NDUFS7* depletion in neurons lowers the failure rate of neurotransmission at very low calcium concentrations.** (**A** and **B**) Representative traces of EPSPs and mEPSPs in (A) motor neuron-Gal4 control (*D42-Gal4/+*) and (B) motor neuron-Gal4 driven *UAS-NDUFS7[RNAi]* (*UAS-NDUFS7[RNAi]/+; D42-Gal4/+*) at 0.15 mM extracellular $Ca^{2+}$ concentration. Scale bars for EPSPs (mEPSP) are $x = 50$ ms (1,000 ms) and $y = 10$ mv (1 mV). (**C**) Quantification of EPSPs in the indicated genotypes. A very low extracellular calcium concentration mildly affects EPSPs in *NDUFS7[RNAi]* compared to control larvae. (**D**) Failure analysis at 0.1 mM $Ca^{2+}$ for the following conditions: Gal4 control (*D42-Gal4/+*), *Sod2* overexpression (*UAS-Sod2/+; D42-Gal4/+*), *D42-Gal4*-driven *UAS-NDUFS7[RNAi]* (*UAS-NDUFS7 [RNAi]/+; D42-Gal4/+*), *UAS-NDUFS7[RNAi]/+; D42-Gal4/+* with NACA, and *UAS-NDUFS7[RNAi]/UAS-Sod2; D42-Gal4/+*. The number of failures was counted per hundred trials in each genotype. There is a marked decrease in failure rate in motor neurons driving *UAS-NDUFS7[RNAi]*. The synaptic failure rates were restored back up to baseline levels

when *UAS-NDUFS7[RNAi]*-depleted flies were reared in the media containing NACA or genetic expression of *Sod2* in motor neurons. (**E** and **F**) Representative paired-pulse EPSP traces at 0.4 mM extracellular $Ca^{2+}$ in the indicated genotypes. Scale bars for EPSPs are x = 50 ms and y = 10 mV. No change in paired-pulse ratio was observed in motor neuron-depleted *UAS-NDUFS7[RNAi]* animals at 0.4 mM $Ca^{2+}$. (G) Quantification of paired-pulse ratio (EPSP2/EPSP1) in motor neuron-Gal4 control (*D42-Gal4/+*) and (B) motor neuron-Gal4 driven *UAS-NDUFS7[RNAi]* (*UAS-NDUFS7 [RNAi]/+; D42-Gal4/+*) larvae. (**H** and **I**) Representative paired-pulse EPSP traces at 1.5 mM extracellular $Ca^{2+}$ in the indicated genotypes. Scale bars for EPSPs are x = 50 ms and y = 10 mV. No change in paired-pulse ratio was observed in motor neuron-depleted UASNDUFS7[RNAi] animals, even at higher calcium concentrations. (**J**) Quantification of paired ratio (EPSP2/EPSP1) in motor neuron-Gal4 control (*D42-Gal4/+*) and (B) motor neuron-Gal4 driven *UAS-NDUFS7[RNAi]* (*UAS-NDUFS7[RNAi]/+; D42-Gal4/+*) larvae at higher extracellular calcium. (**K** and **L**) Representative EPSP recordings of 100 stimuli at 3 mM extracellular $Ca^{2+}$ during a 10 Hz stimulus train in the indicated genotypes. Scale bars for EPSPs are x = 1,000 ms and y = 10 mV. (**M**) Quantifying EPSP amplitudes in the indicated stimulus number in the genotypes mentioned above. (**N** and **O**) Representative high-frequency recordings of 6,000 stimuli at 2 mM extracellular $Ca^{2+}$ during a 10 Hz stimulus train in the indicated genotypes. Scale bars for EPSPs are x = 1,000 ms and y = 10 mV. (**P**) Quantification of the EPSP amplitudes in the indicated stimulus number in the genotypes mentioned above. No significant depletion in vesicle pools was observed when *UAS-NDUFS7[RNAi]*-depleted animals were subjected to high-frequency nerve stimulation for 10 min. *p*-values are indicated in the figure, ns; they are not significant. Statistical analysis based on Student's *t* test. Error bars signify mean ± s.e.m. Raw data for this figure are available in the S2 Data Excel file, tab S11 Fig.
(TIF)

**S12 Fig. Related to Fig 4. Rotenone incubation increases the levels of active zone material at NMJs.** (**A**) Representative images of the A2 hemisegment of muscle 6/7 NMJs in (A) *w^1118^*+DMSO, 1 hour, nerve severed, (**B**) *w^1118^*+DMSO, 1 hour, nerve intact, (**C**) *w^1118^*+500 µM rotenone (ROT), 1 hour, nerve severed, (**D**) *w^1118^*+ROT, 1 hour, nerve intact, (**E**) *w^1118^*+DMSO, 2 hours, nerve severed, (**F**) *w^1118^*+DMSO, 2 hours, nerve intact, (**G**) (*w^1118^*+500 µM ROT, 2 hours, nerve severed), (**H**) (*w^1118^*+ROT, 2 hours, nerve intact), (**I**) (*w^1118^*+DMSO, 4 hours, nerve severed), (**J**) (*w^1118^*+DMSO, 4 hours, nerve intact), (**K**) (*w^1118^*+500 µM ROT, 4 hours, nerve severed), (**L**) (*w^1118^*+ROT, 4 hours, nerve intact), (**M**) (*w^1118^*+DMSO, 6 hours, nerve severed), (**N**) (*w^1118^*+DMSO, 6 hours, nerve intact), (**O**) (*w^1118^*+500 µM ROT, 6 hours, nerve severed), (**P**) (*w^1118^*+500 µM ROT, 6 hours, nerve intact), (**Q**) (*w^1118^*+DMSO, 8 hours feeding, intact larvae), (**R**) (*w^1118^*+25 µM ROT, 8 hours feeding, intact larvae), (**S**) (*w^1118^*+DMSO, embryo to 3rd instar), and (**T**) (*w^1118^*+25 µM ROT, embryo to 3rd instar) larvae immunostained with antibodies against the active zone scaffold Bruchpilot (BRP:fire-LuT) to label the active zones. Scale bar: 5 µm. Note that the incubation or feeding larvae with rotenone elevate BRP levels at the NMJs in *w^1118^*+500 µM ROT, 6 hours nerve intact and *w^1118^*+25 µM ROT, embryo to 3rd instar compared to control nerve severed larvae. (**U**) Histograms show the quantification of BRP intensity in µm² of bouton area at muscle 6/7 in the indicated genotypes. At least 8 NMJs of each genotype were used for quantification. \*\*\**p* = 0.0003 (*w^1118^*+DMSO, 1 hour, nerve severed vs. *w^1118^*+DMSO, 1 hour, nerve intact), \*\*\**p* < 0.0001, \*\**p* = 0.0019 (*w^1118^*+DMSO, embryo to 3rd instar vs. *w^1118^*+25 µM ROT, embryo to 3rd instar). Error bars signify the standard error of the mean. Statistical analysis is based on the Student's *t* test for pairwise comparison among the samples. Raw data for this figure are available in the S2 Data Excel file, tab S12 Fig.
(TIF)

**S13 Fig. Related to Fig 4. Neither *Catalase* nor *Sod1* overexpression reverse synaptic and mitochondrial phenotypes in *NDUFS7*-depleted animals.** Representative images of the A2 hemisegment of muscle 6/7 NMJs in (**A**) *UAS-mitoGFP, D42-Gal4/+*, (**B**) *UAS-Cat/+; UAS-mitoGFP, D42-Gal4/+*, (**C**) *UAS-Sod1/+; UAS-mitoGFP, D42-Gal4/+*, (**D**) *NDUFS7[RNAi]/+; UAS-mitoGFP, D42-Gal4/+*, (**E**) *NDUFS7[RNAi]/UAS-Cat; UAS-mitoGFP, D42-Gal4/+*, and (**F**) *NDUFS7[RNAi]/UAS-Sod1; UAS-mitoGFP, D42-Gal4/+* larvae immunostained with antibodies against the active zone scaffold Bruchpilot (BRP:fire-LuT) to label the active zones. BRP levels are upregulated at the NMJs in *NDUFS7*-depleted

flies, while overexpression of ROS scavenger genes *Cat* or *Sod1* in the neuron fails to restore BRP to the control level. (**A**–**F**) Scale bar: 2.5 μm. (**F** and **G**) Histograms showing quantification of (F) BRP intensity and (G) density per μm² area of bouton at muscle 6/7 for the genotypes mentioned above. At least 8 NMJs of each genotype were used for quantification. ***$p < 0.0001$. Error bars denote mean ± s.e.m. Statistical analysis based on one-way ANOVA followed by post-hoc Tukey's multiple-comparison test. (**H**–**P**) Representative electrophysiological traces and quantifications of mEPSPs, EPSPs and quantal content in the indicated genotypes. Scale bars for EPSPs (mEPSP) are $x = 50$ ms (1,000 ms) and $y = 10$ mV (1 mV). EPSP amplitudes were maintained in *NDUFS7*-depleted flies, likely due to the induction of BRP. A minimum of 7 NMJs recordings of each genotype were used for quantification. mEPSP amplitude: **$p = 0.002$ (Control versus *UAS-Cat*), *$p = 0.009$ (Control versus *UAS-Sod1*), *$p = 0.018$ (Control versus *NDUFS7[RNAi]*); Quantal content: **$p = 0.0009$ (Control versus *UAS-Cat*), *$p = 0.004$ (Control versus *UAS-Sod1*), *$p = 0.010$ (Control versus *NDUFS7[RNAi]*); ns, not significant. Statistical analysis based on one-way ANOVA followed by post-hoc Tukey's multiple-comparison test. Error bars denote the standard error of the mean. (**Q**) Representative images of the A2 hemisegment of muscle 6/7 NMJs in *UAS-mitoGFP, D42-Gal4*/+, (**R**) *UAS-Cat*/+; *UAS-mitoGFP, D42-Gal4*/+, (**S**) *UAS-Sod1*/+; *UAS-mitoGFP, D42-Gal4*/+, (**T**) *NDUFS7[RNAi]*/+; *UAS-mitoGFP, D42-Gal4*/+, (**U**) *NDUFS7[RNAi]*/*UAS-Cat*; *UAS-mitoGFP, D42-Gal4*/+ and (**V**) *NDUFS7[RNAi]*/*UAS-Sod1*; *UAS-mitoGFP, D42-Gal4*/+ larvae immunostained with antibodies against HRP (magenta) and GFP (mito-GFP:green) to label neurons and mitochondria. *NDUFS7*-depleted and *Cat*- or *Sod1*-expressing *NDUFS7[RNAi]* animals harbor fewer mitochondria at the terminals than control animals. (**Q**–**V**) Scale bar: 5 μm. (**W**) Histograms showing quantification of mitochondrial clusters in the above-indicated genotypes. At least 8 NMJs of each genotype were used for quantification. ***$p < 0.0001$. Error bars represent mean ± s.e.m. Statistical analysis based on one-way ANOVA followed by post-hoc Tukey's multiple-comparison test. Raw data for this figure are available in the S2 Data Excel file, tab S13 Fig. (TIF)

**S14 Fig.  Related to Fig 5. MCI regulates levels of active zone material to stabilize synaptic strength in conjunction with intracellular calcium channels.**  Representative images of the A2 hemisegment of muscle 6/7 NMJs in (**A**) *elaV(C155)*/+ with DMSO, (**B**) *elaV(C155)*/+ with Xestospongin C (Xesto) and Dantrolene (Dant), (**C**) *elaV(C155)*/+; *UAS-NDUFS7[RNAi]*/+ with DMSO, (**D**) *elaV(C155)*/ +; *UAS-NDUFS7[RNAi]*/+ with Xesto and Dant, (**E**) *elaV(C155)*/+; *UAS-IP₃-sponge.m30*/+, (**F**) *elaV(C155)*/+; *UAS-mcu[RNAi]*/+, (**G**) *elaV(C155)*/+; *UAS-mcu[RNAi]*/*UAS-NDUFS7[RNAi]* with Xesto and Dant and (**H**) *elaV(C155)*/+; *UAS-mcu[RNAi]*/*UAS-NDUFS7[RNAi]*/+; *UAS-IP₃-sponge.m30*/+ with Dant. Larvae were immunostained with antibodies against the active zone scaffold Bruchpilot (BRP:fire-LuT) to label the active zones. BRP levels are upregulated at the NMJs in *UAS-NDUFS7[RNAi]* with DMSO and *UAS-NDUFS7[RNAi]* with Xesto and Dant flies. However, genetically inhibiting ER calcium release and calcium import to the mitochondria reduces BRP levels below the control level. (**A**–**H**) Scale bar: 5 μm. (**I**) Histograms showing quantification of BRP intensity in the μm2 area of bouton at muscle 6/7 in the above genotypes. At least 8 NMJs of each genotype were used for quantification. ***$p < 0.0001$ **$p = 0.0002$, (BRP levels: *elaV(C155)*/+ with DMSO vs. *elaV(C155)*/+; *UAS-NDUFS7[RNAi]*/+ with Xesto and Dant, Error bars denote mean ± s.e.m. Statistical analysis based on one-way ANOVA followed by post-hoc Tukey's multiple-comparison test. Raw data for this figure are available in the S2 Data Excel file, tab S14 Fig. (TIF)

**S15 Fig.  Related to Fig 6. Neuronal loss of an MCI subunit impairs evoked neurotransmission when combined with loss of glycolysis or TCA cycle genes.**  (**A**–**N**) Representative electrophysiological traces of Gal4 driver control (*elaV(C155)-Gal4*/+) or experimental genotypes tested for combinatorial effects when losing *NDUFS7* gene function and/or the function of glycolysis or TCA cycle genes, including: *hexokinase A* (*hex-A*), *hexokinase C* (*hex-C*), *Citrate (Si) Synthase I*, *Isocitrate dehydrogenase* (*Idh*), and *Succinyl-coenzyme A synthetase α subunit 1* (*Scsa1*). Scale bars for EPSPs (mEPSP) are $x = 50$ ms (1,000 ms) and $y = 10$ mV (1 mV). (**O**–**Q**) Data histograms and statistical analyses for these same genotypes (knockdowns all in the elaV(C155)-Gal4/ + genetic background; see Table Q in the S1 Tables file

for full genotypes), including quantal size (mEPSP), evoked excitation (EPSP) and quantal content. Error bars denote mean ± s.e.m. Statistical analysis based on one-way ANOVA followed by post-hoc Tukey's multiple-comparison test. *$p < 0.05$, **$p < 0.01$, ***$p < 0.001$. Raw data for this figure are available in the S2 Data Excel file, tab S15 Fig. (TIF)

**S16 Fig. Related to Fig 6. Neuronal loss of glycolysis or TCA cycle genes prevents the active zone enhancements induced by *NDUFS7* loss.** (A–L) Representative images of the A2 hemisegment of muscle 6/7 NMJs, examining active zone material through anti-Bruchpilot (BRP) immunostaining. Analysis for Gal4 driver control (*elaV(C155)-Gal4/+*) or experimental genotypes tested for combinatorial effects of losing *NDUFS7* gene function and/or the function of glycolysis or TCA cycle genes, including *hexokinase A* (*hex-A*), *hexokinase C* (*hex-C*), *Citrate (Si) Synthase I*, *Isocitrate dehydrogenase* (*Idh*), and *Succinyl-coenzyme A synthetase α subunit 1* (*Scsa1*). Scale bar: 5 μm. See Table R in the S1 Tables file for full genotypes. (M) Histograms showing quantification of BRP intensity in the μm$^2$ area of bouton at muscle 6/7 in the above genotypes. At least 8 NMJs of each genotype were used for quantification. Error bars signify mean ± s.e.m. Raw data for this figure are available in the S2 Data Excel file, tab S16 Fig. (TIF)

**S17 Fig. Related to Fig 7. NDUFS7 in muscle supports the organization of the α-Spectrin scaffold at the NMJs.** (A–E) Representative confocal images of third instar larval NMJs in (A) Muscle-Gal4 control (*BG57-Gal4/+*), (B) Muscle Gal4-driven *UAS-Sod2* (*UAS-Sod2/+; BG57-Gal4/+*), (C) Muscle *UAS-NDUFS7[RNAi]* (*UAS-NDUFS7[RNAi]/+; BG57-Gal4/+*), (D) Sod2 muscle rescue (*UAS-Sod2/UAS-NDUFS7[RNAi]; BG57-Gal4/BG57-Gal4*), and (E) (*UAS-NDUFS7[RNAi]/+; BG57-Gal4/+* with NACA). Synapses are immunostained with anti-HRP (green) and α-Spectrin (magenta) antibodies. Scale bar: 5 μm. (N) Histogram showing relative α-Spectrin area in the indicated genotypes. Compared with controls, *UAS-NDUFS7[RNAi]/ +; BG57-Gal4/+*) NMJs show a significant reduction in α-Spectrin area, which is restored upon muscle overexpression of a Sod2 transgene or feeding larvae with NACA. ***$p < 0.0001$; ns, not significant. Error bars signify mean ± s.e.m. Statistical analysis based on one-way ANOVA with post-hoc Tukey's test for multiple comparisons. Raw data for this figure are available in the S2 Data Excel file, tab S17 Fig. (TIF)

**S18 Fig. Related to Fig 8. Neither Catalase nor Sod1 overexpression restore defective glutamate receptor-active zone apposition in NDUFS7-deficient animals.** The NDUFS7 subunit in muscle affects the organization of the GluR cluster in Drosophila. Representative confocal images of boutons at third instar larval NMJ synapse in (A) Muscle-Gal4 control (*BG57-Gal4/+*), (B) Muscle Gal4-driven *UAS-Cat* (*UAS-Cat/+; BG57-Gal4/+*), (C) Muscle Gal4-driven *UAS-Sod1* (*UAS-Sod1/+; BG57-Gal4/+*), (D) Muscle *NDUFS7[RNAi]* (*NDUFS7[RNAi]/+; BG57-Gal4/+*), (E) *Cat* muscle rescue (*UAS-Cat/NDUFS7[RNAi]; BG57-Gal4/BG57-Gal4*), and (F) *Sod1* muscle rescue (*UAS-Sod1/NDUFS7[RNAi]; BG57-Gal4/BG57-Gal4*) animals immunolabeled with active zone marker BRP (magenta) and anti-GluRIII (green) antibodies. Scale bar: 2.5 μm. Note that GluRIII clusters apposed by BRP are missing in the *NDUFS7*-depleted NMJs, as well as the *Cat* and *Sod1* non-rescued NMJs (marked in arrow). (G) Histograms showing quantification of the number of missing BRP-GluRIII apposed puncta per bouton in the indicated genotypes. ***$p < 0.0001$. Error bars represent mean ± s.e.m. Statistical analysis based on one-way ANOVA with post-hoc Tukey's test for multiple comparisons. Raw data for this figure are available in the S2 Data Excel file, tab S18 Fig. (TIF)

**S1 Tables. Tables A–U: summary data for graphs in all figures and supplemental figures.** Table V: detailed information about reagents and materials for items described in the STAR methods section. (PDF)

**S1 Data. Excel file with 9 tabs showing raw data for graphs in the main figures.** (XLSX)

**S2 Data. Excel file with 18 tabs showing raw data for graphs in the supplemental figures.**
(XLSX)

## Acknowledgments

We acknowledge the Developmental Studies Hybridoma Bank (University of Iowa, USA) for antibodies used in this study and the Bloomington Drosophila Stock Center (Indiana University, Bloomington, USA) for fly stocks. We thank members of the Frank lab for their helpful comments and discussions in this study.

## Author contributions

**Conceptualization:** Bhagaban Mallik, Sajad A. Bhat, Xinnan Wang, C. Andrew Frank.

**Data curation:** Bhagaban Mallik.

**Formal analysis:** Bhagaban Mallik, Sajad A. Bhat, Xinnan Wang, C. Andrew Frank.

**Funding acquisition:** Xinnan Wang, C. Andrew Frank.

**Investigation:** Bhagaban Mallik, Sajad A. Bhat.

**Methodology:** Bhagaban Mallik, Sajad A. Bhat, Xinnan Wang, C. Andrew Frank.

**Project administration:** C. Andrew Frank.

**Resources:** Xinnan Wang, C. Andrew Frank.

**Supervision:** Xinnan Wang, C. Andrew Frank.

**Validation:** Bhagaban Mallik, Sajad A. Bhat, Xinnan Wang, C. Andrew Frank.

**Visualization:** Bhagaban Mallik.

**Writing – original draft:** Bhagaban Mallik, C. Andrew Frank.

**Writing – review & editing:** Bhagaban Mallik, Sajad A. Bhat, Xinnan Wang, C. Andrew Frank.

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
