## [Editor Report · Decision Letter 0]

10 Feb 2025

Dear Dr Frank, 

Thank you for submitting your manuscript entitled "Mitochondrial Complex I and ROS control synapse function through opposing pre- and postsynaptic mechanisms" for consideration as a Research Article by PLOS Biology.

Your manuscript has now been evaluated by the PLOS Biology editorial staff as well as by an academic editor with relevant expertise and I am writing to let you know that we would like to send your submission out for external peer review.

Once your full submission is complete, your paper will undergo a series of checks in preparation for peer review. After your manuscript has passed the checks it will be sent out for review. To provide the metadata for your submission, please Login to Editorial Manager (https://www.editorialmanager.com/pbiology) within two working days, i.e. by Feb 12 2025 11:59PM.

Kind regards,

Christian

Christian Schnell, PhD

Senior Editor

PLOS Biology

cschnell@plos.org

---

## [Decision Letter · Decision Letter 1]

19 Mar 2025

Dear Dr Frank,

Thank you for your patience while your manuscript "Mitochondrial Complex I and ROS control synapse function through opposing pre- and postsynaptic mechanisms" was peer-reviewed at PLOS Biology. It has now been evaluated by the PLOS Biology editors, an Academic Editor with relevant expertise, and by several independent reviewers. 

In light of the reviews, which you will find at the end of this email, we would like to invite you to revise the work to thoroughly address the reviewers' reports.

As you will see below, the reviewers are largely supportive of your study, but also raise a number of concerns that will need to be addressed. We encourage you to focus in particular on addressing the concerns raised by Reviewer 2 and Reviewer 3, for example showing that the ND20 subunit and the other MCI subunit and other RNAi targets is present in the brain/muscles and that it is knocked down by the RNAi treatment to back up the manipulations. 

We would also like to offer you the opportunity to send us your revision plan so that it is clear to us what work you will try to do up front. This can help avoiding any misunderstanding in the interpretations of these reviewer reports. If you send us your revision plan before your invest time and effort, we can then evaluate it and provide our feedback.

Given the extent of revision needed, we cannot make a decision about publication until we have seen the revised manuscript and your response to the reviewers' comments. Your revised manuscript is likely to be sent for further evaluation by all or a subset of the reviewers.

**IMPORTANT - SUBMITTING YOUR REVISION**

*Re-submission Checklist*

*Published Peer Review*

*PLOS Data Policy*

*Blot and Gel Data Policy*

Sincerely,

Christian

Christian Schnell, PhD

Senior Editor

PLOS Biology

cschnell@plos.org

REVIEWS:

Reviewer #1: The authors report the consequences of suppressing the expression of one of the subunits of Complex I, the first enzyme complex in the electron transport chain, on both mitochondrial, muscle and presynaptic biology in drosophila larvae. On the whole the results appear robust but the logic of the manuscript is somewhat flawed and it makes interpretation difficult. The works reads somewhat like an encyclopedic account of what happened for a particular perturbation without nailing down much in the way of mechanism. The major difficulties in lending interpretation to the findings stem from the following:

There are several consequences one would expect for a mitochondrial perturbation: 1) a bioenergetic consequence, i.e. the sequalae of insufficient production of ATP, 2) related to the above but potentially different, the loss of cataplerotic or increase in anaplerotic metabolites that normally are exported or imported from/into mitochondria; 3) improper Ca handling due to the loss of a major potential Ca buffer, i.e. the mitochondrial transient uptake of Ca via MCU; 4) the consequences of deleterious handling of oxygen within the electron transport chain resulting in an over-abundance of reactive oxygen. A priori one would not, from a pure bioenergetic point of view, expect that the stimulus paradigm used here would be a particularly stringent test of a bioenergetic compromise that one might anticipate from a mitochondrial perturbation. Perhaps more problematic is that the authors are using unrealistic concentration of the main consumable external carbon source, in this case 115 mM sucrose, perhaps 2 orders of magnitude too high. Unfortunately, it is very hard to understand what functional impact anything might mean when probing function under these metabolic conditions and with relatively sparsely demands (as in the case of Fig 1) or even during repetitive stimulation (Fig S6)- the latter though has additionally modified another parameter, external Ca. With respect to Ca handling the authors have made use of a few tools to interrupt contributions of ER stores, but in the absence of some functional measure of Ca in either the ER, the mitochondrial matrix or the cytoplasm (ideally all 3) , these are not easy to interpret. The other major complexity arises from the fact that although ROS scavenging (or the lack there of) has impact on presynaptic biology, it is very unclear if any of this is actually a local effect or one that arises in the cell body. As even the chemical scavenging perturbation is done over the time scale of development (actually not precisely stated but I am inferring this from the statement "grown in" in the methods). Thus, although conditions that lead to excess ROS production appear to drive interesting changes in presynaptic organization, whether this arose locally from axonal or presynaptic mitochondria or was a transcriptional response in the cell body is very unclear. In this regard it could simply be that the deleterious ROS impact in muscles is also confined to the cell body, but given that this is not spatially very separate from the endplate means that the consequences are different that in neurons but the reason may be more spatial than anything else.

At this stage this manuscript is not ready for publication without trying to attack these points in a logical fashion. I am surprised the authors did not try to make better use of tools like rotenone, a bona fide Complex I inhibitor. Comparisons of the consequence of more acute pharmacological intervention compared to a chronic genetic one might lead to greater insight and interpretability of the results. 

Reviewer #2 (Sean Sweeney): Mallik and Frank have produced and extensive and wide-ranging study of ROS, mitochondrial function, calcium regulation at the pre- and post-synaptic compartment. While manipulating mitochondrial complex I (MCI) in the presynaptic compartment, they identify changes in mitochondrial shape, networks and distribution. Changes in microtubule structures and increases in pre-synaptic active zones are also observed and these are rescued via expression of mitochondrially localised SOD2. With MCI manipulation there is no overt change to the synaptic output, but a change is revealed when the manipulation is rescued with SOD2 expression. In the presence of MCI knockdown, manipulations of ER Ca2+ (inhibition) reveal deficits in EPSP that are not present unless MCI is knocked down. When reading, I was then asking what ATP levels looked like, but the authors took a different but complementary route to check this: Glycolysis also appears to play an active role, in the absence of EPSP deficits when MCI is knocked down, also regulating active zone levels. 

The study then switches to muscles and the manipulation of MCI there. Synapse overgrowth, decrease in EPSP, loss of the PSD95 protein (dlg) and GluR puncta are observed, and critically some of these aspects can be rescued by SOD2 expression or NACA feeding, but not SOD1 (approximates to cytoplasmic ROS) or catalase. 

There is much more, and a lot to digest in this paper, but I feel it is pretty comprehensive. I would have liked to see if the rescues with SOD1 or catalase also failed to rescue the pre-synaptic phenotypes, and if these attempts also rescued GluRIII loss. However, I feel the paper would stat to become unwieldy and the authors have made some very important novel points regarding the actions of ROS in the nervous system. The paper is very clearly written and easy to follow (tho the volume of critical data is large!), the images and graphs clear. 

I have a couple of minor suggestions, typos: 

Figure 1, L and M, the shape of the nuclei in M (I assume they are nuclei?) suggests this part of the prep is a bit squished, perhaps improve image M? Or is nuclear shape part of the phenotype? If so, please mention this in the text. 

Fig 4, is there an internal control for the brp staining for quantification? 

Line 700 Cystine should be Cysteine. 

Line 711, due to induction of brp??

Fig 5, increase the size of the green arrows and red crosses to make the figures work - they are the operative symbols and need to dominate a little more. 

In the main text - please give a rationale for the switch between the OK371-GAL4 driver to the Elav-GAL4 driver. The switch needs to be explained. 

Images aren't quite as sharp in fig 8 - use a higher res image? 

The paper is very well referenced, but still there are two papers I would suggest that would be helpful: Joe Bateman's previous study showing MCI knockdown and and the resulting pre-synaptic effect on the nervous system in flies: doi: 10.1371/journal.pgen.1010793

and more for the discussion, a recent Stephan Sigrist paper (DOI: 10.1016/j.redox.2024.103454) showing a similar accumulation of active zone material with age (i.e. why we study ROS?) in the adult brain. 

Reviewer #3: This is an interesting paper authored by Mallik and Frank. Here, they describe how knocking down ND-20L, a CI subunit, increases mitochondrial superoxide, affecting synapse function and the Drosophila neuromuscular junction (NMJ). The study targets both the presynaptic (motor neurons) and the postsynaptic (muscle) compartments, yielding contrasting effects.

When ND-20L is targeted in motor neurons, ROS levels increase, triggering a homeostatic compensatory mechanism to maintain presynaptic excitation. However, when ND-20L is targeted in muscle cells, ROS levels also rise, but muscle cells are disrupted, leading to synapse degeneration and alterations in neurotransmission.

The authors utilised a diverse repertoire of Drosophila reagents, including RNAi and the overexpression of various genes, to manipulate CI and superoxide levels, mitochondrial fission and fusion, and other relevant processes. Additionally, they employed several pharmacological interventions using ROS scavengers such as N-Acetyl L-cysteine amide (NACA) and inhibitors like rotenone, as well as drugs affecting calcium signalling and glycolysis. A broad range of technical approaches was used, including live imaging to detect ROS using MitoSOX, electrophysiological recordings, immunocytochemistry, and confocal microscopy to visualise and quantify presynaptic active zone proteins, postsynaptic scaffolding proteins, glutamate receptors, and mitochondrial morphology.

The manuscript is well written, with detailed materials and methods and well-presented data. However, I have several concerns that must be addressed, as the current version presents serious issues regarding the validity of its conclusions. I want to be clear: if the following key issues are addressed with experimental data—either using previously published/unpublished data or new experiments—I will accept the conclusions of the paper without hesitation.

Major Comments

1. According to FlyBase, ND-20L is a testis-specific subunit, with no or minimal expression in all other adult and larval tissues, except the imaginal disc in L3. Therefore, it is difficult to justify how CI could be depleted in motor neurons or muscle cells, or how the observed phenotypes could result from CI depletion.

The short-hairpin sequence of the RNAi line used matches only ND-20L, as expected, and not ND-20, indicating that it should not deplete the non-specific testis form, ND-20.

Given this, the authors must provide evidence of ND-20L expression at significant levels in the studied tissues (nervous system, muscle) and demonstrate CI depletion at the protein level. This can be done by measuring mitochondrial oxygen consumption, complex I activity, or performing Blue Native Gel Electrophoresis (BNE). I would prefer the first two methods, as they are quantitative, but I would accept BNE.

If obtaining sufficient material from larval muscle or nervous tissue is not feasible, an alternative—albeit less ideal—approach would be to measure CI activity in larval homogenates or adult flies using a ubiquitous driver (see PMID: 38304969, PMID: 27076081, PMID: 34977670). These experiments are essential to validate the model and determine the extent of CI depletion achieved by targeting ND-20L.

In my opinion, depletion should be negligible, which would invalidate the model, but I acknowledge that I may be mistaken. Thus, these experiments would be highly informative. If such data have been previously generated by the same laboratory (I could not find them in the previous Frontiers publication), they should be clearly referenced. I want to emphasise that the results must be unambiguous, as the consensus is that testis-specific CI subunits are only expressed in the testis. If this is not the case, a new consensus must be reached.

Alternatively, all experiments could be repeated using ND-30. I don't understand why the authors didn't use ND-30 instead of ND-20L. However, similar to the previous point, I would like to see evidence that this RNAi line effectively depletes CI in larvae. These kinds of experiments are important for validating the model.

2. The efficiency of the other RNAi lines and the overexpression levels of genes such as Sod2 must be verified. This can be assessed using a ubiquitous driver and larval homogenates. While not a perfect method, it would at least provide proof that the RNAi lines function as intended. I have not seen such verification in the present manuscript. Again, references to previous studies—preferably from the same laboratory but always using the same RNAi lines—would be acceptable.

These experiments, particularly (1), are critical, as the results presented in the manuscript contradict what is currently known about ND-20L. Therefore, it must be demonstrated that the model functions as proposed.

3. The authors should use mitochondrially targeted catalase and repeat key experiments. They used NACA and SOD2, which have very different mechanisms of action and are localised in different cellular compartments.

SOD2 detoxifies superoxide in the mitochondrial matrix.

NACA provides reductive power to the antioxidant system and can detoxify radicals directly, but it is unclear whether it accumulates at sufficiently high concentrations in the mitochondrial matrix to be effective.

Since other antioxidants, such as SOD1 or catalase, do not work, and SOD2 does—reducing superoxide but increasing H₂O₂—this suggests that superoxide produced in the matrix is responsible for the observed phenotypes. However, if that is the case, how does NACA rescue the phenotype?

MitoSox experiments in Figure S1 show that NACA reduces matrix superoxide (the MitoSox signal), further supporting the idea that matrix superoxide is the key ROS. This is mechanistically relevant, as superoxide is charged and cannot leave the mitochondria. The mitochondrially targeted catalase (available at the Bloomington Drosophila Stock Center) experiment would confirm whether the effect is specifically due to superoxide.

If mitochondrial catalase does not affect the MitoSox signal and fails to rescue the phenotype, it would confirm that superoxide is the key ROS.

These experiments should be performed in both motor neurons and muscle.

I consider these experiments essential. However, I understand that, given the extensive work already performed, the authors may prefer to state that mitochondrial superoxide is likely the cause of the observed effects but that the results remain inconclusive. If so, they should explicitly acknowledge the study's limitations and indicate that the mitochondrial catalase experiment (or overexpressing a peroxiredoxin in the mitochondria) would clarify the specific role superoxide or hydrogen peroxide involved.

4. Knocking down Sod2 would further clarify the role of superoxide. If Sod2 knockdown increases the MitoSox signal, it should produce the same effects as knocking down ND-20L.

If it does not increase the MitoSox signal, it would suggest that without CI depletion, superoxide levels remain very low.

Again, I consider this an excellent experiment, but I understand that the authors may have a different perspective and prefer to discuss point 3 instead.

Additional Concerns

5. The authors should specify whether they used males, females, or a mix of both. If they did not distinguish between sexes, this must be explicitly stated in the Materials and Methods. Again, ND-20L should not be expressed in females, but I am open to data that proves otherwise. If such data have already been published, they need to be referenced.

6. Were the cultures synchronised? If not, how do the authors ensure that control and experimental animals are at the same developmental stage? Knocking down mitochondrial genes can cause developmental delay.

7. Using red and green in figures can make interpretation difficult for colour-blind readers. Consider modifying the colour scheme to improve accessibility.

Minor Comments

1. Page 3: A reference should be added for the total number of CI subunits (42 in this paper), as different sources report varying numbers.

2. Page 3: I believed the 14 core catalytic subunits are conserved in Drosophila. Am I mistaken?

3. Page 3: The literature on CI depletion and ROS shows conflicting results (PMID: 36794623, PMID: 3535522, PMID: 24243023). These discrepancies should be discussed.

4. Figure 3, panels E-E': The images do not appear representative of a rescue phenotype. Do the authors have a better example?

5. The authors mention three different temperatures, (18, 25 and 29) in Materials and Methods, but these are not referenced again. Were these different temperatures used to control gene expression?

---

## [Editor Report · Decision Letter 2]

12 Aug 2025

Dear Andy,

Thank you for your patience while we considered your revised manuscript "Mitochondrial Complex I and ROS control synapse function through opposing pre- and postsynaptic mechanisms" for publication as a Research Article at PLOS Biology. This revised version of your manuscript has been evaluated by the PLOS Biology editors and the Academic Editor.

Based on our Academic Editor's assessment of your revision, we are likely to accept this manuscript for publication, provided you satisfactorily address the following data and other policy-related requests:

* We would like to suggest a different title to improve its accessibility for our broad audience: 

"Mitochondrial Complex I and ROS control synapse function through opposing pre- and post-synaptic mechanisms at neuromuscular junctions"

* Please add the links to the funding agencies in the Financial Disclosure statement in the manuscript details.

DATA POLICY:

Regardless of the method selected, please ensure that you provide the individual numerical values that underlie the summary data displayed in the following figure panels as they are essential for readers to assess your analysis and to reproduce it: 1KPS, 2CDM, 3F, 4FGNOPU, 5QRST, 6HIJQ, 7JKLUV, 8FK, 9EFGH, S1ABCDE, S2IJ, S3CFIL, S4CD, S5CD, S6MNO, S7L, S8M, S9G, S10GHI, S11CDGJ, S12U, S13GNOPW, S14I, S15OPQ, S16M, S17F and S18G.

* CODE POLICY

We expect to receive your revised manuscript within two weeks. 

*Published Peer Review History*

*Press*

Sincerely,

Christian

Christian Schnell, PhD 

Senior Editor

cschnell@plos.org

PLOS Biology

---

## [Editor Report · Decision Letter 3]

28 Aug 2025

Dear Andy,

Thank you for the submission of your revised Research Article "Mitochondrial Complex I and ROS control neuromuscular function through opposing pre- and postsynaptic mechanisms" for publication in PLOS Biology. On behalf of my colleagues and the Academic Editor, Josh Dubnau, I am pleased to say that we can in principle accept your manuscript for publication, provided you address any remaining formatting and reporting issues. These will be detailed in an email you should receive within 2-3 business days from our colleagues in the journal operations team; no action is required from you until then. Please note that we will not be able to formally accept your manuscript and schedule it for publication until you have completed any requested changes.

PRESS

We frequently collaborate with press offices. If your institution or institutions have a press office, please notify them about your upcoming paper at this point, to enable them to help maximize its impact. If the press office is planning to promote your findings, we would be grateful if they could coordinate with biologypress@plos.org. If you have previously opted in to the early version process, we ask that you notify us immediately of any press plans so that we may opt out on your behalf.

Sincerely, 

Christian

Christian Schnell, PhD

Senior Editor

PLOS Biology

cschnell@plos.org